# VTool-R1: VLMs Learn to Think with Images via Reinforcement Learning on Multimodal Tool Use

**Mingyuan Wu**[1][*]**, Jingcheng Yang**[1][*]**, Jize Jiang**[1]**, Meitang Li**[2]**, Kaizhuo Yan**[1]**,**
**Hanchao Yu**[3]**, Minjia Zhang**[1]**, Chengxiang Zhai**[1]**, Klara Nahrstedt**[1]

[1]University of Illinois Urbana-Champaign
[2]University of Michigan Ann Arbor, [3]Independent Researcher
{mw34, klara}@cs.illinois.edu

## ABSTRACT

Reinforcement learning finetuning (RFT) has significantly advanced the reasoning capabilities of large language models (LLMs) by enabling long chains of thought, multi-turn self-correction, and effective tool use. While recent works attempt to extend RFT to vision-language models (VLMs), these efforts largely focus on text-only reasoning conditioned on original image inputs, and do not incorporate visual reasoning in the response. In contrast, test-time methods like Visual Sketchpad incorporate visual steps but lack training mechanisms.

We introduce **VTool-R1**, the first RFT framework that trains VLMs to generate multimodal chains of thought by interleaving text and intermediate visual reasoning steps. **VTool-R1** integrates Python-based visual editing tools into the RFT process, enabling VLMs to learn when and how to generate visual reasoning steps that enhance the final output quality. Trained with outcome-based rewards, our approach elicits strategic visual tool use for multi-modal reasoning without relying on process-based supervision. Extensive experiments on structured visual reasoning over charts and tables show that **VTool-R1** enhances reasoning performance by teaching VLMs to "think with images" and generate multimodal chain of thoughts with tools. To support future research in multi-turn multi-modal reasoning, we open-source our code at https://github.com/VTOOL-R1/vtool-r1.

## 1 INTRODUCTION

Recent large language models (LLMs), notably DeepSeekR1 (DeepSeek-AI, 2025) and the GPT4o series (OpenAI, 2024), have demonstrated remarkable capabilities in text-based reasoning. Central to this advancement is Reinforcement learning finetuning (RFT), which enables these models to learn to generate long chains of thought and engage in self-correction and verification for complex reasoning tasks (Kumar et al., 2025; Zeng et al., 2025). Beyond boosting intrinsic reasoning capabilities, RFT has also shown promising results in effectively integrating external tool use, such as search engines (Jin et al., 2025; Chen et al., 2025b) and code interpreters (Feng et al., 2025), into the reasoning process of LLMs.

Despite this rapid progress in LLM reasoning brought by RFT, there has not yet been a well-recognized breakthrough in improving multimodal reasoning capabilities of vision-language models (VLMs). Modern VLMs (Bai et al., 2025) typically consist of a strongly language-aligned image encoder that maps visual inputs into feature space, then a connector module further projects these visual features into the token space of a decoder-only LLM. On top of these architectures, prior works have attempted to replicate post-training strategies first developed on LLMs, like visual instruction tuning (Xu et al., 2024), to improve VLMs' ability to follow textual instructions.

However, similar attempts by recent works (Zhou et al., 2025; Chen et al., 2025a; Zhang et al., 2025; Huang et al., 2025; Liu et al., 2025; Deng et al., 2025) to replicate the success of RFT on

---

[*]Mingyuan and Jingcheng contributed equally.

LLMs onto the VLM domain to enhance multimodal reasoning fall short: these reasoning approaches remain fundamentally **text-driven**: they handle images only during the initial encoding stage and generate reasoning chains purely in text conditioned on fixed input image tokens, without involving any intermediate visual reasoning steps.

**Why is text-dominant reasoning insufficient?** Even the most advanced VLMs may rely on language shortcuts if text dominates reasoning. For example, consider a prompt where a user shows GPT-5 (OpenAI, 2025) an image of a hand with six fingers and asks, "How many fingers are in this hand?" The model may respond with "five" with high chance, based on the pure text reasoning path —"a hand has five fingers". This illustrates a critical failure of purely language-based reasoning in multimodal settings: the model can decode a more plausible answer through wrong textual chain of thoughts shortcuts.

To go beyond text-based reasoning, early inference frameworks like Visual Sketchpad (Hu et al., 2024) and Refocus (Fu et al., 2025) demonstrate that incorporating intermediate visual reasoning steps during inference can improve multimodal reasoning performance. However, these inference-time methods rely on the usage of highly capable models such as GPT-4o to produce meaningful visual steps, and underperform when used with open-source, less capable models.

In this paper, we present the first RFT frameworks that directly enables VLMs to learn to think in a combination of images and texts and to be trained to generate multimodal chains of thoughts. Our framework, called **VTool-R1**, teaches VLMs to generate intermediate visual reasoning steps through interactions with external image editing tools implemented in Python code. These visual reasoning steps are interleaved with textual chains of thought, resulting in multi-turn multimodal reasoning steps during the generation of the response. Following the design of DeepSeekR1, VTool-R1 uses only outcome-based rewards—the model is not explicitly rewarded for generating visual steps, but instead learns when and how to incorporate visual reasoning steps in order to improve final task performance.

We demonstrate the effectiveness of VTool-R1 on challenging structured image reasoning tasks, focusing on table and chart-based reasoning. Our experiments use a well-curated dataset and a visual editing tool set from Refocus. This toolset, which is directly callable by the VLM, enables selective attention on relevant regions in the table and chart - simulating how humans process visual information through attention before forming final reasoning results. Through VTool-R1, the model learns to use these tools strategically to guide its multimodal chain of thought and enhance final reasoning performance.

Our key contributions can be summarized as follows:

- We present VTool-R1, a novel RFT framework that supports VLM multimodal reasoning with external visual editing tool use. We demonstrate that RFT with outcome based reward design can incentive generation of visual reasoning steps for final reasoning accuracy.

- To the best of our knowledge, our work is the first work that successfully enables VLMs to learn to integrate intermediate visual reasoning steps into text-based chains of thoughts in the generated response (i.e. thinking with a combination of images and text).

- We validate our approach through extensive controlled experiments on challenging and well-established structured image reasoning datasets, using a predefined visual toolset in Refocus dataset.

## 2 RELATED WORKS

### 2.1 VISUAL CHAIN OF THOUGHT REASONING

Early works have demonstrated that incorporating visual intermediate steps—often generated via external tools or Python scripts—can benefit a wide range of visual question answering (VQA) tasks, without any training. Pioneering efforts such as ViperGPT (Surís et al., 2023) and Visual Programming without Training (Gupta & Kembhavi, 2023) utilize Python-based visual tools to manipulate images during inference. Cache of Thought (Wu et al., 2025) further improves multimodal reasoning by retrieving high-quality rationales from similar queries. Recently, Visual Sketchpad (Hu et al., 2024) introduced a framework that equips multimodal language models with a sketchpad and drawing

tools. The model is prompted to generate visual artifacts during inference and uses them for iterative planning and reasoning. While this approach successfully introduces visual information into the reasoning process, it operates solely at inference time, with no training involved. Refocus (Fu et al., 2025) takes a step forward by prompting the VLM to invoke visual editing tools for selective attention over images. These modified images are then used as inputs for further reasoning. However, Refocus does not train the model to reason with tools; instead, it relies on pre-edited images generated by a commercially powerful model such as GPT-4o, and hence underperforms in scenarios where only less-performant, smaller models are available.

## 2.2 LLM/VLM, REINFORCEMENT LEARNING AND TOOL USE

The trend of RFT for LLMs began with RLHF (Ouyang et al., 2022), using PPO (Schulman et al., 2017), and later evolved into more efficient variants like DPO (Rafailov et al., 2023) and SimPO (Meng et al., 2024), though these often face off-policy limitations. GRPO (Shao et al., 2024) mitigates such issues with a critic-free, group-based design that improves both efficiency and stability. RFT has also been used to guide tool-augmented reasoning in LLMs (Chen et al., 2025b; Feng et al., 2025; Jin et al., 2025), demonstrating that outcome-based rewards can teach models when and how to use tools during multi-step reasoning.

In the VLM domain, recent works attempt to adapt RFT for VLMs to incentivize multimodal reasoning behaviors (Zhou et al., 2025; Chen et al., 2025a; Zhang et al., 2025; Huang et al., 2025; Liu et al., 2025; Deng et al., 2025; Wang et al., 2025), but they primarily train VLMs to only generate textual chains of thought from visual inputs. However, training VLMs to produce and reason over intermediate visual steps via tool use remains largely unexplored. Concurrent works related to VTool-R1, such as Deepeyes (Zheng et al., 2025) and OpenThink-IMG (Su et al., 2025), also integrate visual steps into reasoning chains, with different tool and task designs. We further demonstrate that VTool-R1 significantly outperforms Deepeyes on structured image data, which we attribute to the intrinsic design of our tools and tasks.

## 3 VTOOL-R1

**VLM Preliminaries**. A VLM policy can be denoted as $\pi_\theta$ parametrized with model weights $\theta$. Given a text prompt sequence $x$ and an image $I$, the model can generate a text response sequence $y$, sampled from the $\pi_\theta(I, x)$. Some VLMs support multiple image inputs; however, their capabilities in parsing and understanding multiple images vary significantly and are highly dependent on the training procedure (Wang et al., 2024). In our setting, we require VLMs capable of processing multiple images as both the original input image and intermediate visual steps must be processed together as input to the VLMs.

In the following sections, we present the detailed design of VTool-R1, covering inference and training parts. In the Section 3.1, we show that how pre-trained VLM can be prompted to use visual editing tools and generate integrate intermediate visual steps. We then go beyond inference in the Section 3.2: by extending the RFT objective, we train VLMs to use these tools and generate multimodal chains of thought during rollout by themselves. We also introduce an outcome-based reward formulation that encourages effective visual reasoning while avoiding the pitfalls of process-based reward hacking.

## 3.1 VLM INFERENCE AND ROLLOUT WITH VISUAL CHAIN OF THOUGHTS

Refocus (Fu et al., 2025) has shown that capable VLMs, such as GPT-4o, can be prompted to generate Python code for calling external tools to edit the input image, followed by reasoning over the resulting visual outputs. Our inference and rollout prompting closely follows their format; full prompt templates are included in the Appendix.

```
VLM Inference and Rollout Prompt

<Image Input><System Prompt><Python Codes Templates>
# GOAL #:  Based on the above tools, I want you to reason about how to solve the # USER
REQUEST # and generate the actions step by step (each action is a python function call)
to solve the request.  You may need to use the tools above to process the images and
```

```
make decisions based on the visual outputs of the previous code blocks.  You should
only use the tools above, you should not use other functions or code that will not be
executed.
# REQUIREMENTS #:
...3.  If you think you got the answer, use ANSWER: <your answer> Please extract the
final answer in FINAL ANSWER: <final answer> and ends with TERMINATE.
...8.  If you do not think you have enough information to answer the question on the
images returned by the tools, you should directly answer the question based on the
original image...
9.  Only one turn of action, ACTION 0, is allowed.  You must provide the answer after a
maximum one ACTION call.
In-context Examples:
<Thought 0><Action 0><Observation><Edited Image><Thought 1><Answer>
# USER Bounding Box Info:  x_values_bbox, storing x values and coordinates.
y_values_bbox, storing x values and coordinates.  The x values in the image are:
<x_values>.  The y values in the image are:  <y_values>.
# USER IMAGE stored in image_1, as PIL image.
```

As illustrated in the prompt box above, we provide system instructions and task-specific goals to guide the VLM in using visual tools for reasoning. The model is given the names and definitions of visual editing functions, along with detailed descriptions of their usage. Through in-context examples, the VLM is prompted to begin its reasoning in *Thought 0*, which outlines where to focus in the image. It then produces *Action 0*, which is either a "no action needed" statement or a Python-like pseudocode snippet that invokes an appropriate visual editing tool.

The tool call is executed externally in a Python environment to generate a modified image, which is then fed back into the model as additional input for an additional turn. The VLM continues its reasoning over this generated intermediate visual step, forming a richer visual chain of thought that supports the final answer. As specified in the Requirement section of the prompt, the model is allowed to either respond directly or invoke a tool once to create an intermediate visual step for further reasoning. In this work, we focus on single-turn tool use—i.e., the model may call a tool at most once and reason over the edited image. Extending this to multi-turn tool use, where the model iteratively edits and reasons for multiple rounds, is left for the future, as effective multi-turn design requires stronger VLMs capable of handling multi-image inputs.

This inference and rollout process is inherently iterative and cannot be completed in a single-turn VLM call if the model chooses to use tools. Execution must pause for the external Python environment to run the generated code. Once the modified image is available, it has to be reintroduced into the same VLM instance as the second image input, rather than being inserted at the original location in the generated response sequence, as is common in prior tool-use approaches such as Search-R1 (Jin et al., 2025).

Formally, if the VLM policy $\pi_\theta$ decides to invoke a tool from an external visual editing toolset $\mathtt{T}$, the inference involves two rounds of model execution. The first round samples an initial response containing tool calls, $y' \sim \pi_\theta(\cdot \mid I, x)$, where the input text prompt $x$ includes tool descriptions. The tool calls are then executed in the Python environment as $I' = \mathtt{T}(y', I)$ to generate a modified image. In the second round, the VLM performs reasoning over both the original and edited images:

$$y \sim \pi_\theta(\cdot \mid I, x; \mathtt{T}) = \pi_\theta(\cdot \mid I \oplus I', x) \\ = \pi_\theta(\cdot \mid I \oplus \mathtt{T}(y', I), x) \tag{1}$$

where $\bigoplus$ denotes the concatenation of the original image $I$ and the updated image $I'$ as dual image inputs to the model. If the VLM chooses not to invoke any tools and instead answers the question directly, the final answer is obtained in the first round without needing a second inference pass: $y \sim \pi_\theta(\cdot \mid I, x)$.

## 3.2 RFT VLM TO GENERATE VISUAL CHAIN OF THOUGHTS

VTool-R1 adopts RFT to train VLMs to explore flexible reasoning trajectories and learn to invoke visual editing tools effectively. Given the two-stage iterative inference structure, two training objective designs are possible—for instance, optimizing only the final response $y$ that yields the answer, or jointly optimizing both the intermediate tool-invoking output $y'$ and the final response $y$. In this work,

we choose to optimize only the final response $y$, as our goal is to directly train the model to improve final reasoning quality. We maintain the notation introduced at the end of Section 3.1.

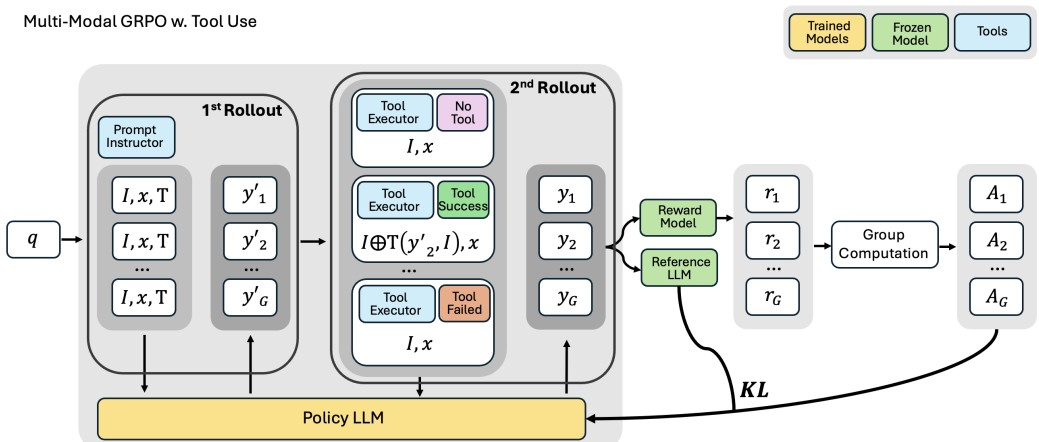

Figure 1: Multi-Modal GRPO w. Tool Use Training Pipeline, where the input $q$ is a multimodal query

In VTool-R1, we assume a reward model $r_\phi$ under the Bradley-Terry formulation, and consider the following RFT training objectives:

*Optimize the final reasoning response $y$ during RL rollout*:

$$\max_{\pi_\theta} \mathbb{E}_{[I,x]\sim\mathcal{D},\, y\sim\pi_\theta(\cdot|I,x;\texttt{T})}\big[r_\phi(I,x,y)] \;-\; \beta\,\mathbb{D}_{\mathrm{KL}}\big[\pi_\theta(\cdot \mid I,x;\texttt{T}) \,\|\, \pi_{\mathrm{ref}}(\cdot \mid I,x;\texttt{T})\big]. \quad (2)$$

where $\pi_\theta$ is the policy VLM parametrized with model weights $\theta$. $\pi_{\mathrm{ref}}$ is the reference VLM policy. $r_\phi$ is the reward function. $\mathbb{D}_{\mathrm{KL}}$ is the KL-divergence measure. $\beta > 0$ is the KL penalty coefficient. The input $[I,x]$ denotes multimodal samples drawn from the dataset $\mathcal{D}$. The generated response in the rollout $y \sim \pi_\theta(\cdot \mid I,x;\texttt{T}) = \pi_\theta(\cdot \mid I \bigoplus I', x) = \pi_\theta(\cdot \mid I \bigoplus \texttt{T}(y',I), x)$, if the model chooses to use a tool; otherwise, when no tool is invoked, the response simplifies to $y \sim \pi_\theta(\cdot \mid I,x;\texttt{T}) = \pi_\theta(\cdot \mid I,x)$.

Unlike prior RFT that simply relies on LLM/VLM policy to generate rollout during training (Ouyang et al., 2022), VTool-R1 explicitly incorporates the visual editing tool use from the toolset $\texttt{T}$ in the rollout, and conditions the model on the edited image input. We refer readers to Section 3.1 for the formal definition and intuition of the iterative tool-use rollout policy, which mirrors the model inference pipeline. This iterative pipeline enables more effective step-by-step reasoning across both modalities: the model learns to modify the image using tools to support its reasoning before producing the final answer.

Note that we do not directly optimize the intermediate tool-invoking response $y'$ in our RFT process, as our goal is to encourage the model to autonomously decide whether using a tool improves reasoning. This design supports a more end-to-end training objective.

Our training approach is built upon a well-established policy gradient method: Group Relative Policy Optimization (GRPO) in (DeepSeek-AI, 2025; Shao et al., 2024), which offers improved stability and eliminates the need for a separate critic model. Unlike Proximal Policy Optimization, which estimates advantages using a learned critic, GRPO estimates the baseline from a group of sampled responses and reduces training resources. Specifically, for each input $[I,x]$, GRPO samples a group of responses $\{y_i\}_{i=1}^{G} \sim \pi_{\mathrm{old}}(\cdot|I,x;T)$ from the old policy model, and then optimize the current policy model by maximizing the following objective equation 3:

Here, $\epsilon$ and $\beta$ are hyper-parameters, and $\hat{A}_{i,t} = \tilde{r}_i = \frac{r_i - \mathrm{mean}(r)}{\mathrm{std}(r)}$ denotes the normalized relative advantage computed within the group of sampled responses. This formulation avoids the need for critic model, while maintaining stable and reward-aligned policy updates by regularizing with the KL divergence between the updated policy $\pi_\theta$ and the reference policy $\pi_{\mathrm{ref}}$.

$$\mathcal{J}_{GRPO}(\theta) = \mathbb{E}_{[I,x]\sim\mathcal{D},\{y_i\}_{i=1}^{G}\sim\pi_{\text{old}}(\cdot|I,x;T)} \tag{3}$$

$$\left[ \frac{1}{G}\sum_{i=1}^{G}\frac{1}{|y_i|}\sum_{t=1}^{|y_i|}\min\Big(r_{i,t}(\theta)\hat{A}_{i,t}, \text{clip}\big(r_{i,t}(\theta), 1-\epsilon, 1+\epsilon\big)\hat{A}_{i,t}\Big) - \beta\mathbb{D}_{KL}\left[\pi_\theta||\pi_{\text{ref}}\right] \right]$$

We use inference template in the Section 3.1 for training rollouts as well. This template structures the model's output, think before actions and let the model decide whether we need a tool call, with the system instructions and requirements. We make the template highly formatted and listed clearly thoughts, actions, tool use function blocks and final answers. We also include few shot examples for better instruction and format following.

### 3.3 REWARD MODELING

Following Deepseek-R1, we adopt an outcome-based reward design that relies solely on the correctness of the model's final answer. For closed-ended tasks like factual QA, an exact string match works well. However, in our setting—structured visual understanding—the answers are more free-form and not easily judged by string match. To address this, we use a lightweight LLM-based judge to merely assess the match between the predicted answer and the ground truth. While not strictly rule-based, this serves as a pseudo rule-based reward appropriate for open-ended tasks such as ChartQA. We reward score of 1 when the judge thinks it is a match.

We also study process-based rewards that penalize incorrect tool use or reward successful invocations. However, this often results in reward hacking in RFT: models either avoid tools entirely when penalties are applied, or exploit success criteria by generating tool calls that superficially meet expectations without contributing to reasoning.

We do not use format-based rewards, as our models already learn to follow the structured format—Thoughts, Actions, Tools, and Final Answer—thanks to clear instruction templates. We leave further exploration of more dedicated format rewards to future work, but find our current setup sufficient for reliable rollout behavior.

## 4 EXPERIMENT

### 4.1 DATASET

Following Refocus (Fu et al., 2025), we evaluate **VTool-R1** on structured image reasoning tasks that are particularly suitable for visual editing tool use. Our evaluation focuses on chart and table-based reasoning questions, which poses significant challenges for early VLM works (Liu et al., 2023; 2022). To ensure fair comparison, **we strictly adhere to the data splits created in Refocus.** These splits are constructed from the following sources:

**Table Split.** We evaluate on three datasets: (1) VWTQ (Pasupat & Liang, 2015), 750 QA pairs from Wikipedia tables rendered as screenshots with HTML; (2) VWTQ_syn (Kim et al., 2024), 250 synthetic tables with randomized styles to avoid training data leakage; and (3) VTabFact (Chen et al., 2019), 250 entailment classification pairs rendered from TabFact content. All are split 70/30 for train/test, with no validation set due to limited size.

**Chart Split.** From ChartQA (Masry et al., 2022) with human written questions grounded in real world charts, Refocus select 444 horizontal and 382 vertical bar chart QA pairs for testing. For training, Refocus collect 14,344 examples and 813 validation examples.

### 4.2 VISUAL EDITING TOOLSET T

While our long-term goal is to enable models to invoke arbitrary tools or APIs within a sandbox environment using outcome-based rewards, we begin by demonstrating VTool-R1's capabilities with a set of simple but effective visual editing tools. These tools are implemented in Python and help simulate visual attention by modifying table or chart images. We adopt the same tool set used in Refocus. In our experiments for tabular problems, we utilize a variety of tools as follows:

*Highlight Column/Row*: overlays a semi-transparent red on the selected columns/Rows.

*Mask Column/Row*: applies a white mask over irrelevant columns/rows.

*Draw Column/Row*: draws a solid red bounding box around selected columns/rows.

For charts, we apply analogous operations to highlight or mask individual bars, based on their positions along the x-axis or y-axis.

The model is instructed to call one or multiple tools at the same time, as many operations (e.g., drawing bounding boxes on multiple positions) can be performed in parallel and we involve at most one round of tool call in the experiments.

These tools leverage external libraries such as OpenCV to perform tasks like drawing bounding boxes and identifying maskable regions based on contours of bars or tables for selective attention in our tasks. Looking ahead, we envision integrating more advanced generative models as powerful tools that can execute more generalized visual modifications directly from language prompts.

## 4.3 EXPERIMENT SETUP

We demonstrate the effectiveness of our RFT framework by training the state-of-the-art open-source VLMs, Qwen-VL 2.5 models (Bai et al., 2025) at 3B, 7B, and 32B scales. Training uses the open-source VeRL training infrastructure (Sheng et al., 2024). Training is conducted with the AdamW optimizer (Loshchilov & Hutter, 2019), using an initial learning rate of $1e-6$ and a weight decay of $1e-2$. Due to the large image sizes and long prompt sequences (up to 16,384 tokens), we set the global batch size to 32 for table split and 256 for chart split. The rollout group size is set to 5, and the KL divergence coefficient to $1e-2$. The 3B/7B models are trained on 8/16 H100 GPUs with 96GB memory; the 32B models are trained on 8 H200 GPUs with 141GB memory. We standardize decoding across rollout and evaluation with temperature 1.0 and bf16 precision.

## 4.4 BASELINE MODELS

| Qwen2.5-VL | 3B | | | 7B | | | 32B | | |
|---|---|---|---|---|---|---|---|---|---|
| | Pure Run | Tool Use | VTool-R1 | Pure Run | Tool Use | VTool-R1 | Pure Run | Tool Use | VTool-R1 |
| Chart Split | 51.8 | 24.6 | 64.0 | 76.2 | 53.4 | 80.7 | 88.0 | 85.0 | 86.7 |
| Table Split | 41.3 | 24.3 | 57.9 | 64.7 | 41.1 | 71.7 | 86.2 | 76.0 | 84.5 |

Table 1: Main Results of VTool-R1 and Baselines in Accuracy

| | VTool-R1 | | R1-VL | | GPT-4o | |
|---|---|---|---|---|---|---|
| | 3B | 7B | 2B | 7B | Pure Run | Tool Use |
| Chart Split | 64.0 | 80.7 | 10.4 | 63.8 | 82.9 | 80.5 |
| Table Split | 57.9 | 71.7 | 8.6 | 45.4 | 75.7 | 77.0 |

Table 2: VTool-R1 compared to other post-RL reasoning models

We report several baselines to contextualize VTool-R1's performance in Table 1 and Table 2:

**GPT-4o (OpenAI, 2024):** Used as an upper bound across all benchmarks. This is a powerful commercial model that has already shown remarkable inference-only tool use capability for reasoning in Refocus (Fu et al., 2025).

**Qwen2.5-VL (3B / 7B / 32B) (Bai et al., 2025):** Included without any RFT in two configurations:

**Prompted Tool-Use Inference**: The model is prompted to use tools following our inference and rollout template in Section 3.1. However, prior to RFT, these open-source models before our RL fine-tuning struggle to follow tool-use instructions. For fairness, when a tool call fails, we append a prompt indicating the failure and ask the model to regenerate its answer.

**Direct Inference (No Tools):** Only the image and question are provided, with no tool-related prompts. Surprisingly, this setting of open-source model yields strong results across datasets—often outperforming even GPT-4o—especially for larger models like 32B.

These findings about high pure run accuracy suggest that Qwen2.5-VL 3B and 7B may have been post-trained on VQA-style tasks or distilled from larger models, enabling strong direct-answering performance, but lacking the general-purpose tool-use capabilities seen in models like GPT-4o. This aligns with observations from Refocus, which found that only commercial models could be prompted to use tools for generating meaningful intermediate visual steps, while open-source models lacked this ability—until the introduction of our VTool-R1 framework.

**Weaker Baseline:** We include R1-VL (Zhang et al., 2025), an open-source RL-tuned model trained on general visual reasoning tasks, as a weaker academic baseline.

## 4.5 MAIN RFT RESULTS AND FINDINGS

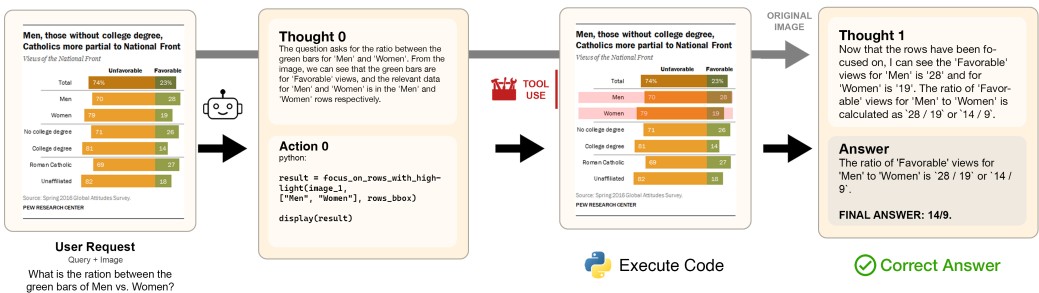

Figure 2: Illustrative Example from VTool-R1 (3B): After RFT, 3B Model Successfully Integrates Intermediate Visual Steps.

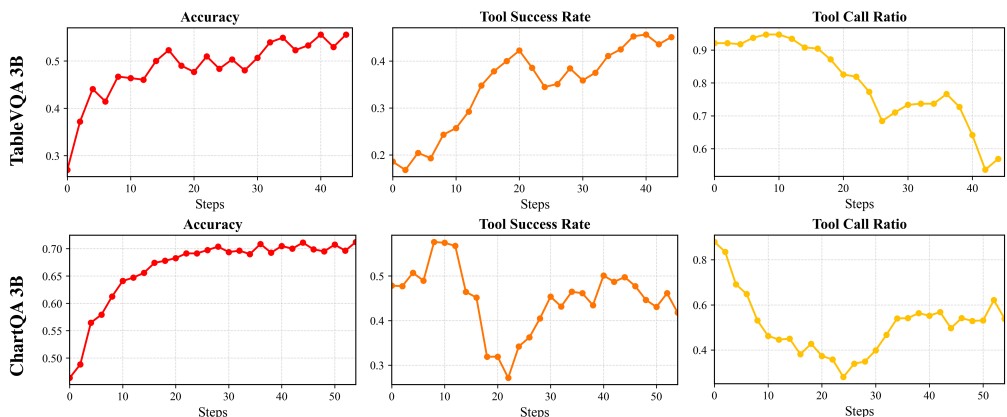

Figure 3: Multi-Modal GRPO w. Tool Use Training Dynamics, for 3B models

**Qualitative Tool-use Example.** Figure 2 presents a qualitative example where the VTool-R1 3B model successfully integrates intermediate visual steps through tool use as part of the reasoning process, ultimately arriving at the correct answer.

**RFT makes Better Tool Use for Reasoning**. When comparing VTool R1 Model reasoning accuracy with non-trained baselines in Table 1, we observe a significant improvement in tool use capability. After training, models learn to reason correctly with multimodal tool use, guided solely by outcome-based rewards. Remarkably, the 3B and 7B model, which initially failed with high chance to generate meaningful tool use (thoughts, actions, or tool calls), learns to use tools effectively to generate intermediate reasoning. This indicates that VTool-R1 not only improves final answer accuracy, but

also enables models to internalize structured reasoning patterns. In both the 3B and 7B cases, **VTool-R1 significantly outperforms the direct inference baseline, highlighting that RFT-driven tool use as an additional reasoning step brings substantial benefits to the model's overall reasoning capabilities.**

**Better Tool Use, or Not? It's Not Monotonic.** Our goal goes beyond improving accuracy—we aim to teach the model when and how to use tools in a way that meaningfully supports reasoning in the RL training. As shown in Figure 3, VTool-R1 enables models to make nuanced, context-aware tool-use decisions. Interestingly, tool call frequency and success rate do not increase monotonically when the training proceeds and accuracy goes up. Instead, we observe fluctuations: models tend to overuse tools early in training due to prompt instruction exposure but later learn to invoke them more selectively. The 3B model becomes more cautious with tool use over time, leading to higher reasoning accuracy. Crucially, the model also learns when tools are unnecessary and confidently proceeds with direct reasoning. The 32B model (training curve shown in the Appendix) exhibits a higher overall tool use rate but similarly shows periods of decline, reflecting adaptive behavior. This adaptive tool-use behavior for reasoning is a key outcome of our RFT strategy. While the exact trends vary between table and chart tasks, the overall pattern remains consistent.

**More Successful Tool Use or Not?** Figure 3 also presents the tool call success rates of the most representative 3B model during RFT training on both chart and table tasks. It is important to note that we cannot evaluate tool call correctness with full precision and recall, as no oracle verifier is available. Instead, we rely on a proxy metric to approximate success: A tool call is considered successful if the python commands executed did not raise any exceptions inside a sandbox environment with the given functions, and a valid pillow image is returned from the execution through passing the processed image into the *display* function. According to this metric, the success rate of tool use steadily increases on table tasks, while for charts it fluctuates throughout training. We would like to highlight the need for future work to incorporate human-annotated oracle verifier to more accurately evaluate tool-use success.

**Training Dynamics**. Overall, the model's accuracy steadily improves throughout training, with minor fluctuations. Performance gradually converges and stabilizes around the final accuracy within approximately 50 training steps in around one epoch. The saturated step number varies depending on the training configuration.

**Reward Design**. We also explore alternative reward settings beyond the standard 0/1 outcome-based reward. When applying process-based rewards, such as penalizing failed tool calls, we observe that the model quickly learns to avoid tool use entirely, driving the tool usage rate to zero. Conversely, when we add extra reward for successful tool use that leads to a correct answer, the model begins to exploit the verifier and hack to triggering a "success" signal. This holds even under stricter verifier criteria. These findings support our claims that outcome-based rewards tied solely to final task correctness serve as the most reliable and robust reward design for VTool-R1.

**Benefits from External Tool Use.** As shown in Table 2, VTool-R1 shows clear advantages over R1-VL models trained on general visual reasoning data, by incorporating an additional tool-use step to generate intermediate visual outputs. These results underscore the impact of learning to "think with images": the added visual reasoning step significantly enhances final performance.

**Experiments v.s. Deepeyes**. Additionally, we present results comparing our approach against a concurrent RL-trained tool-use VLM, Deepeyes. Our 7B model trained with VTool-R1 achieves significantly higher performance. 60.0 (deepeyes) v.s. 80.7 (ours), which we attribute to the intrinsic good design of our tools, training recipe, and well-designed structured image tasks.

**Failure Analysis**, In particular, we highlight several representative categories of failure cases, including: (a). The model correctly generates an intermediate reasoning visual step, but fails to make the correct inference in the second rollout. (b). The model correctly identified the intermediate visual step needed for the reasoning, but the augmentation is slightly flawed (numbers obstructed by the added bounding box). The second rollout fails to read the correct numbers from the edited image. (in Appendix) (c). The model concludes in the first rollout that no image reasoning step is not needed, but fails to provide the correct answer. (in Appendix) (d). The model generated tool use code for but the code execution was unsuccessful, and no intermediate visual step is provided. (in Appendix)

---

**Failure Case #1**

Query: What is the value of largest bar?

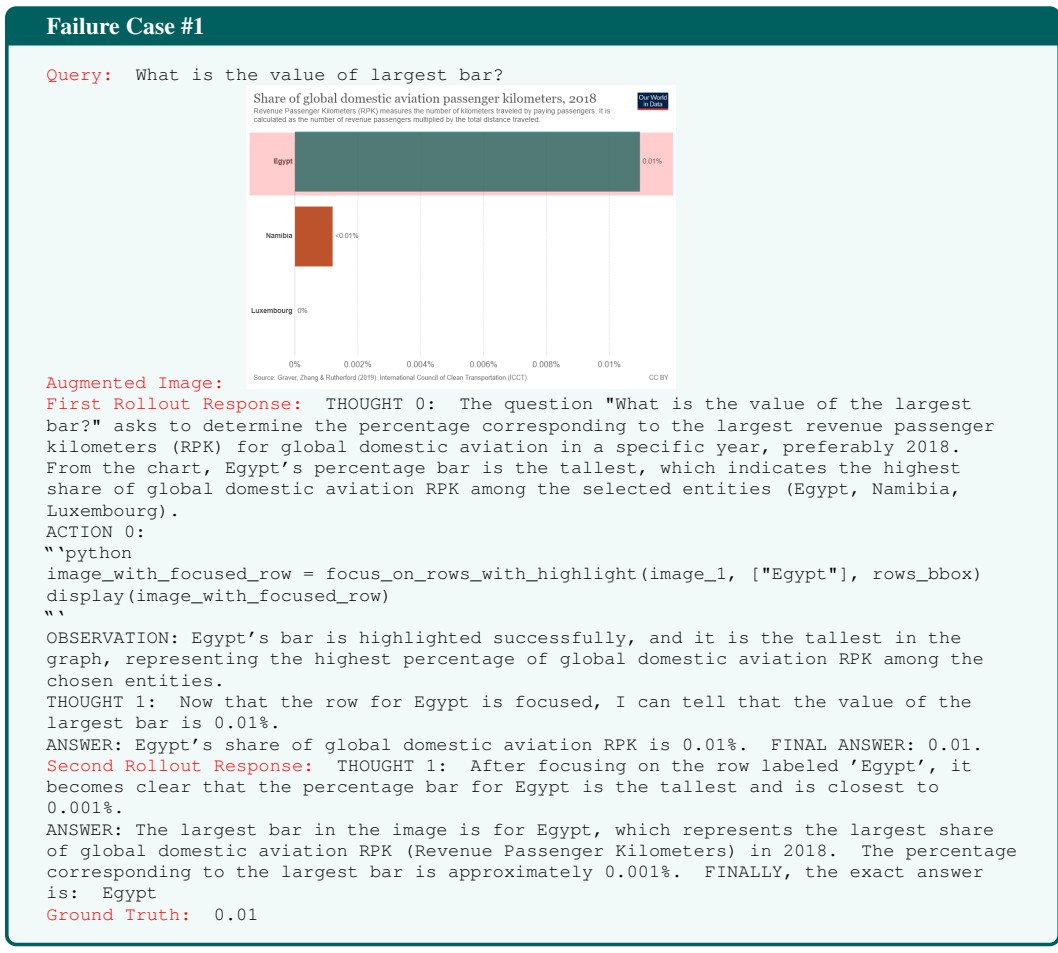

Augmented Image:
First Rollout Response: THOUGHT 0: The question "What is the value of the largest bar?" asks to determine the percentage corresponding to the largest revenue passenger kilometers (RPK) for global domestic aviation in a specific year, preferably 2018. From the chart, Egypt's percentage bar is the tallest, which indicates the highest share of global domestic aviation RPK among the selected entities (Egypt, Namibia, Luxembourg).
ACTION 0:
```python
image_with_focused_row = focus_on_rows_with_highlight(image_1, ["Egypt"], rows_bbox)
display(image_with_focused_row)
```
OBSERVATION: Egypt's bar is highlighted successfully, and it is the tallest in the graph, representing the highest percentage of global domestic aviation RPK among the chosen entities.
THOUGHT 1: Now that the row for Egypt is focused, I can tell that the value of the largest bar is 0.01%.
ANSWER: Egypt's share of global domestic aviation RPK is 0.01%. FINAL ANSWER: 0.01.
Second Rollout Response: THOUGHT 1: After focusing on the row labeled 'Egypt', it becomes clear that the percentage bar for Egypt is the tallest and is closest to 0.001%.
ANSWER: The largest bar in the image is for Egypt, which represents the largest share of global domestic aviation RPK (Revenue Passenger Kilometers) in 2018. The percentage corresponding to the largest bar is approximately 0.001%. FINALLY, the exact answer is: Egypt
Ground Truth: 0.01

---

## 5 CONCLUSION

VTool-R1 demonstrates that RFT can effectively teach VLMs to reason by interleaving textual and visual steps. By integrating visual editing tools into the RL training loop and optimizing for outcome-based rewards, VTool-R1 enables models to learn when and how to use tools to support their reasoning, without requiring process-level supervision. Extensive experiments on structured visual reasoning tasks show that VTool-R1 not only improves final reasoning task accuracy, but also equips models to generate coherent, multimodal chains of thought with intermediate reasoning steps in the image modality.

**Broader Impacts, Limitations and Future Directions**. VTool-R1 is one of the first frameworks to show that RFT can train VLMs to integrate visual reasoning steps by invoking visual editing tools and generating intermediate visual states to support their own reasoning goals. This opens up a novel and promising direction for multimodal AI, enabling models to reason more effectively across modalities and potentially unlocking fundamentally new capabilities that go beyond what is encoded in model parameters, especially for more tasks. VTool-R1 holds strong potential for scaling to a broader range of toolsets and generalizing to more diverse datasets. However, in this work, we focus on a straightforward task, selective attention in structured image reasoning, as a starting point. Looking forward, we expect future extensions of the VTool-R1 framework to support multi-turn execution over more sophisticated and diverse tools. Beyond external tool calls, future multi-turn post-training framework could also incorporate model internal feedback (Liao et al., 2025; Wu et al., 2026) to further advance multimodal reasoning and agentic systems.

## 6 ACKNOWLEDGEMENTS

This research used the Delta advanced computing and data resource which is supported by the National Science Foundation (award OAC 2005572) and the State of Illinois. Delta is a joint effort of the University of Illinois Urbana-Champaign and its National Center for Supercomputing Applications.

We would also like to acknowledge Bowen Jin (author of Search-R1) and Xingyu Fu (author of Refocus) for their valuable suggestions and contributions to our project.

This work was supported by the National Science Foundation grants NSF 2106592, NSF 1900875, and NSF 2441601. Any results and opinions are our own and do not represent views of National Science Foundation.

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

# A  APPENDIX

**LLM Usage**: We use LLM for grammar checks and writing polish.

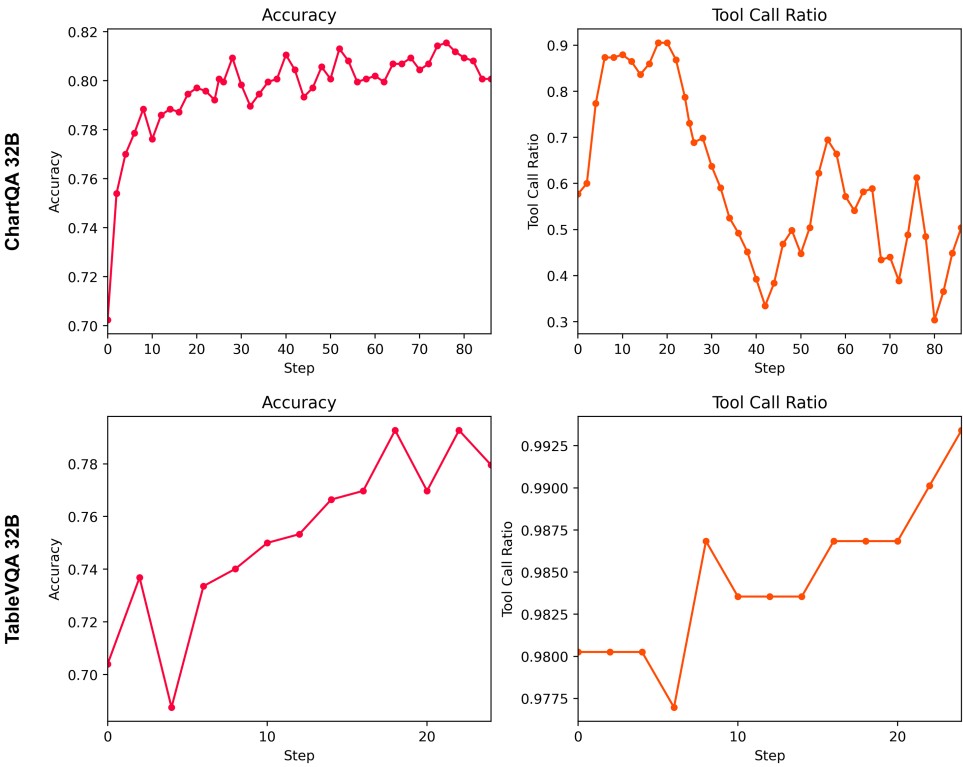

Figure 4: Multi-Modal GRPO w. Tool Use Training Dynamics, for 32B models

---

**Prompts**

```
Here are some tools that can help you.  All are python codes.
They are in tools.py and will be imported for you.  You will
be given a table figure:  image_1 and a question.  Notice
that you, as an AI assistant, are not good at answering
questions when there are too many unnecessary and irrelevant
information.  You should determine which are the relevant
columns to the question, and specify them in a python list.
You should use the given column headers.  You should also
determine which are the relevant rows to the question, and
specify them in a python list.  You should use the given row
headers.  You could select the tools to focus on some columns
/ rows, or mask out some columns / rows.  Use whichever tool
you think is more appropriate.  Below are the tools in
tools.py:

```python
def focus_on_columns_with_highlight(image,
    columns_to_focus_on, all_columns_bounding_boxes):
    \"\"\"
    This function is useful when you want to focus on some
    specific columns of the image.
```

```
    It does this by adding light transparent red highlight to
    the columns that need to be focused on.
   For example, you can focus on the columns in a table that
    are relevant to your analysis.
   Return the drawed image.

   Args:
       image (PIL.Image.Image): the input image
       columns_to_mask (List[str]): a list of column names
   to focus on.
       all_columns_bounding_boxes (Dict[Dict]]): a
   dictionary of bounding boxes for all columns in the image
   . key is column name and value is the bounding box of
   that column. Each bounding box is in the format {'x1': x1
   , 'y1': y1, 'x2': x2, 'y2': y2}.

   Returns:
       image_with_focused_columns (PIL.Image.Image): the
   image with specified columns focused on

   Example:
       image = Image.open("sample_img.jpg")
       image_with_focused_columns =
   focus_on_columns_with_highlight(image, ["Year", "Name"],
   {"Year": {'x1': 0.1, 'y1': 0.1, 'x2': 0.3, 'y2': 0.9}, "
   Team": {'x1': 0.4, 'y1': 0.1, 'x2': 0.6, 'y2': 0.9}, "
   Name": {'x1': 0.7, 'y1': 0.1, 'x2': 0.9, 'y2': 0.9}})
       display(image_with_focused_columns)
   \"\"\"

def focus_on_rows_with_highlight(image, rows_to_focus_on,
   all_rows_bounding_boxes):
   \"\"\"
   This function is useful when you want to focus on some
   specific rows of the image.
   It does this by adding light transparent red highlight to
    the rows that need to be focused on.
   For example, you can focus on the rows in a table that
   are relevant to your analysis.
   Return the drawed image.

   Args:
       image (PIL.Image.Image): the input image
       rows_to_focus_on (List[str]): a list of row headers
   to focus on.
       all_rows_bounding_boxes (Dict[Dict]): a dictionary of
    bounding boxes for all rows in the image. key is row
   header and value is the bounding box of that row. Each
   bounding box is in the format {'x1': x1, 'y1': y1, 'x2':
   x2, 'y2': y2}.

   Returns:
       image_with_focused_rows (PIL.Image.Image): the image
   with specified rows focused on

   Example:
       image = Image.open("sample_img.jpg")
```

```
        image_with_focused_rows =
    focus_on_rows_with_highlight(image, ["1972"], ["Year": {'
    x1': 0.1, 'y1': 0.1, 'x2': 0.9, 'y2': 0.15}, "1969": {'x1
    ': 0.1, 'y1': 0.2, 'x2': 0.9, 'y2': 0.5}, "1972": {'x1':
    0.1, 'y1': 0.6, 'x2': 0.9, 'y2': 0.9}])
        display(image_with_focused_rows)
    \"\"\"

def focus_on_columns_with_mask(image, columns_to_focus_on,
    all_columns_bounding_boxes):
    \"\"\"
    This function is useful when you want to focus on some
    specific columns of the image.
    It does this by masking out the columns that are not
    needed.
    For example, you can focus on the columns in a table that
    are relevant to your analysis and ignore the rest.
    Return the masked image.

    Args:
        image (PIL.Image.Image): the input image
        columns_to_mask (List[str]): a list of column names
    to focus on.
        all_columns_bounding_boxes (Dict[Dict]]): a
    dictionary of bounding boxes for all columns in the image
    . key is column name and value is the bounding box of
    that column. Each bounding box is in the format {'x1': x1
    , 'y1': y1, 'x2': x2, 'y2': y2}.

    Returns:
        image_with_focused_columns (PIL.Image.Image): the
    image with specified columns focused on

    Example:
        image = Image.open("sample_img.jpg")
        image_with_focused_columns = focus_on_columns(image,
    ["Year", "Name"], {"Year": {'x1': 0.1, 'y1': 0.1, 'x2':
    0.3, 'y2': 0.9}, "Team": {'x1': 0.4, 'y1': 0.1, 'x2':
    0.6, 'y2': 0.9}, "Name": {'x1': 0.7, 'y1': 0.1, 'x2':
    0.9, 'y2': 0.9}})
        display(image_with_focused_columns)
    \"\"\"

def focus_on_rows_with_mask(image, rows_to_focus_on,
    all_rows_bounding_boxes):
    \"\"\"
    This function is useful when you want to focus on some
    specific rows of the image.
    It does this by masking out the rows that are not needed.
    For example, you can focus on the rows in a table that
    are relevant to your analysis and ignore the rest.
    Return the masked image.

    Args:
        image (PIL.Image.Image): the input image
        rows_to_focus_on (List[str]): a list of row headers
    to focus on.
```

```
        all_rows_bounding_boxes (Dict[Dict]): a dictionary of
     bounding boxes for all rows in the image. key is row
    header and value is the bounding box of that row. Each
    bounding box is in the format {'x1': x1, 'y1': y1, 'x2':
    x2, 'y2': y2}.

     Returns:
        image_with_focused_rows (PIL.Image.Image): the image
    with specified rows focused on

     Example:
        image = Image.open("sample_img.jpg")
        image_with_focused_rows = focus_on_rows(image,
    ["1972"], ["Year": {'x1': 0.1, 'y1': 0.1, 'x2': 0.9, 'y2
    ': 0.15}, "1969": {'x1': 0.1, 'y1': 0.2, 'x2': 0.9, 'y2':
     0.5}, "1972": {'x1': 0.1, 'y1': 0.6, 'x2': 0.9, 'y2':
    0.9}])
        display(image_with_focused_rows)
    \"\"\"

def focus_on_columns_with_draw(image, columns_to_focus_on,
    all_columns_bounding_boxes):
    \"\"\"
    This function is useful when you want to focus on some
    specific columns of the image.
    It does this by drawing a red box around the columns that
     need to be focused on.
    For example, you can focus on the columns in a table that
     are relevant to your analysis.
    Return the drawed image.

    Args:
        image (PIL.Image.Image): the input image
        columns_to_mask (List[str]): a list of column names
    to focus on.
        all_columns_bounding_boxes (Dict[Dict]): a
    dictionary of bounding boxes for all columns in the image
    . key is column name and value is the bounding box of
    that column. Each bounding box is in the format {'x1': x1
    , 'y1': y1, 'x2': x2, 'y2': y2}.

     Returns:
        image_with_focused_columns (PIL.Image.Image): the
    image with specified columns focused on

     Example:
        image = Image.open("sample_img.jpg")
        image_with_focused_columns = focus_on_columns(image,
    ["Year", "Name"], {"Year": {'x1': 0.1, 'y1': 0.1, 'x2':
    0.3, 'y2': 0.9}, "Team": {'x1': 0.4, 'y1': 0.1, 'x2':
    0.6, 'y2': 0.9}, "Name": {'x1': 0.7, 'y1': 0.1, 'x2':
    0.9, 'y2': 0.9}})
        display(image_with_focused_columns)
    \"\"\"

def focus_on_rows_with_draw(image, rows_to_focus_on,
    all_rows_bounding_boxes):
```

```
    \"\"\"
    This function is useful when you want to focus on some
    specific rows of the image.
    It does this by drawing a red box around the rows that
    need to be focused on.
    For example, you can focus on the rows in a table that
    are relevant to your analysis.
    Return the drawed image.

    Args:
        image (PIL.Image.Image): the input image
        rows_to_focus_on (List[str]): a list of row headers
    to focus on.
        all_rows_bounding_boxes (Dict[Dict]): a dictionary of
     bounding boxes for all rows in the image. key is row
    header and value is the bounding box of that row. Each
    bounding box is in the format {'x1': x1, 'y1': y1, 'x2':
    x2, 'y2': y2}.

    Returns:
        image_with_focused_rows (PIL.Image.Image): the image
    with specified rows focused on

    Example:
        image = Image.open("sample_img.jpg")
        image_with_focused_rows =
    focus_on_columns_with_highlight(image, ["1972"], ["Year":
     {'x1': 0.1, 'y1': 0.1, 'x2': 0.9, 'y2': 0.15}, "1969":
    {'x1': 0.1, 'y1': 0.2, 'x2': 0.9, 'y2': 0.5}, "1972": {'
    x1': 0.1, 'y1': 0.6, 'x2': 0.9, 'y2': 0.9}])
        display(image_with_focused_rows)
    \"\"\"
```

# GOAL #:  Based on the above tools, I want you to reason
about how to solve the # USER REQUEST # and generate the
actions step by step (each action is a python function call)
to solve the request.  You may need to use the tools above to
process the images and make decisions based on the visual
outputs of the previous code blocks.  You should only use the
tools above, you should not use other functions or code which
will not be executed.
# REQUIREMENTS #:
1.  The generated actions can resolve the given user request
# USER REQUEST # perfectly.  The user request is reasonable
and can be solved.  Try your best to solve the request.
2.  The arguments of a tool must be the same format specified
in # TOOL LIST #;
3.  If you think you got the answer, use ANSWER: <your
answer> Please extract the final answer in FINAL ANSWER:
<final answer> and ends with TERMINATE.
4.  All images in the initial user request are stored in PIL
Image objects named image_1, image_2, ..., image_n.  You can
use these images in your code blocks.  Use display() function
to show the image in the notebook for you too see.
5.  Use as few tools as possible.  Only use the tools for the
use cases written in the tool description.  You can use
multiple tools in a single action.

```
6.  If you have multiple answers, please separate them with
|| marks.  For example, if the answer is 'Alice' and 'Bob',
you should write 'Alice||Bob'.
7.  When you focus on columns in the image, most like you
need to look at multiple columns instead of a single one.
8.  If you do not think you have enough information to answer
the question on the images returned by the tools, you should
directly answer the question based on the original image.
Below are some examples of how to use the tools to solve the
user requests.  You can refer to them for help.  You can also
refer to the tool descriptions for more information.
9.  Only one turn of action, ACTION 0, is allowed.  You must
provide the answer after maximum one ACTION call.

# EXAMPLE: Simple question that does not require any tool
# USER REQUEST #:  <A image here> What is the title of this
table?
# USER Bounding Box Info:  columns_bbox, where keys are
column headers and values are column bounding boxes.
rows_bbox, where keys are row headers and values are row
bounding boxes.  The columns in the image are:  ["Grade",
"Mentor", "Salary"].  The rows in the image start with:
["Grade", "A", "B", "C"].
# USER IMAGE stored in image_1, as PIL image.
# RESULT #:
THOUGHT 0:  The question does not require any tool.  I can
see the title of the table is "Customer Information".
ACTION 0:  No action needed.
ANSWER: The title of the table is "Customer Information".
FINAL ANSWER: Customer Information.  TERMINATE

# EXAMPLE: Focus on specific columns in the image
# USER REQUEST #:  <A image here> Who had the same game
version as John Roth?
# USER Bounding Box Info:  columns_bbox, where keys are
column headers and values are column bounding boxes.
rows_bbox, where keys are row headers and values are row
bounding boxes.  The columns in the image are:  ['Manager
Name', 'Game Version', 'Game Score'].  The rows in the image
start with:  ['Manager Name', 'John Roth', 'Alice Smith',
'Bob Johnson'].
# USER IMAGE stored in image_1, as PIL image.
# RESULT #:
THOUGHT 0:  To identify who had the same game version as John
Roth, I need to focus on the 'Game Version' column and the
'Manager Name' column.  Also, I need to focus on all the rows
so I do not need to focus on some specific rows.
ACTION 0:

```python
image_with_focused_columns = focus_on_columns_with_draw(image
    \_1, ["Game Version", "Manager Name"], columns\_bbox)
display(image_with_focused_columns)
```

OBSERVATION: Execution success.  The output is as follows:
<the image outputs of the code is added as the second image>
THOUGHT 1:  Now I can see the 'Game Version' column and the
'Manager Name' column more clearly.  The game version of John
```

Roth is 'v1.2'. Other people with the same game version are 'Alice Smith' and 'Bob Johnson'.
ANSWER: 'Alice Smith' and 'Bob Johnson' had the same game version as John Roth are. FINAL ANSWER: Alice Smith||Bob Johnson. TERMINATE

```python
# EXAMPLE: Focus on specific rows in the image
# USER REQUEST #: <A image here> How many games did Josh win after 1996?
# USER Bounding Box Info: columns_bbox, where keys are column names and values are column bounding boxes. rows_bbox, where keys are row headers and values are row bounding boxes. The columns in the image are: ["Rank", "Year", "Score", "Month"]. The rows in the image start with: ["Rank", "0", "1", "2", "3", "4", "5"].
# USER IMAGE stored in image_1, as PIL image.
# RESULT #:
THOUGHT 0: This table is about the games played by Josh that he won. I do not need to focus on any specific columns. I need to focus on the rows with the year after 1996. Three rows have year after 1996: one has year 1997 and this row starts with "3", one has year 1998 and this row starts with "4", and one has year 1999 and this row starts with "5". So I will focus on the rows with row starters "3", "4", and "5".
ACTION 0:

```python
image_with_focused_rows = focus_on_rows_with_highlight(image\
    _1, ["3", "4", "5"], rows\_bbox)
display(image_with_focused_rows)
```

OBSERVATION: Execution success. The output is as follows:
<the image outputs of the code is added as the second image>
THOUGHT 1: I can see that Josh won 3 games after 1996.
ANSWER: Josh won 3 games after 1996. FINAL ANSWER: 3.
TERMINATE

# EXAMPLE: Focus on specific columns and specific rows in the image
# USER REQUEST #: <A image here> what is the sum of annual earnings after 2006?
# USER Bounding Box Info: columns_bbox where keys are column names and values are column bounding boxes. rows_bbox, where keys are row headers and values are row bounding boxes. The columns in the image are: ["Index", "Year", "Cost", "Earning"]. The rows in the image start with: ["Index", "0", "1", "2", "3", "4", "5"].
# USER IMAGE stored in image_1, as PIL image.
# RESULT #:
THOUGHT 0: I need to focus on the 'Year' column and the 'Earning' column. I also need to focus on the rows with years after 2006. The row with year being 2006 starts with "3". So I will focus on the rows starting with "4", and "5".
ACTION 0:

```python
image_with_focused_columns = focus_on_columns_with_mask(image\
    \_1, ["Year", "Earning"], columns\_bbox)
```

```
image_with_focused_rows = focus_on_rows_with_draw(
    image_with_focused_columns, ["4", "5"], rows\_bbox)
display(image_with_focused_rows)
```

OBSERVATION: Execution success.  The output is as follows:
<the image outputs of the code is added as the second image>
THOUGHT 1:  I can see that the annual earnings after 2006 are
$165,498 and $198,765.  The sum of the annual earnings after
2006 is $364,263.
ANSWER: The sum of the annual earnings after 2006 is
$364,263.  FINAL ANSWER: 364263.  TERMINATE.
# USER Bounding Box Info:  x_values_bbox, storing x values
and coordinates.  y_values_bbox, storing x values and
coordinates.  The x values in the image are:  <x_values>.
The y values in the image are:  <y_values>.
# USER IMAGE stored in image_1, as PIL image.

2. The model correctly identified the augmentation needed for the problem, but the augmentation is slightly flawed (numbers obstructed by the added bounding box). The second rollout fails to read the correct numbers from the augmented graph.

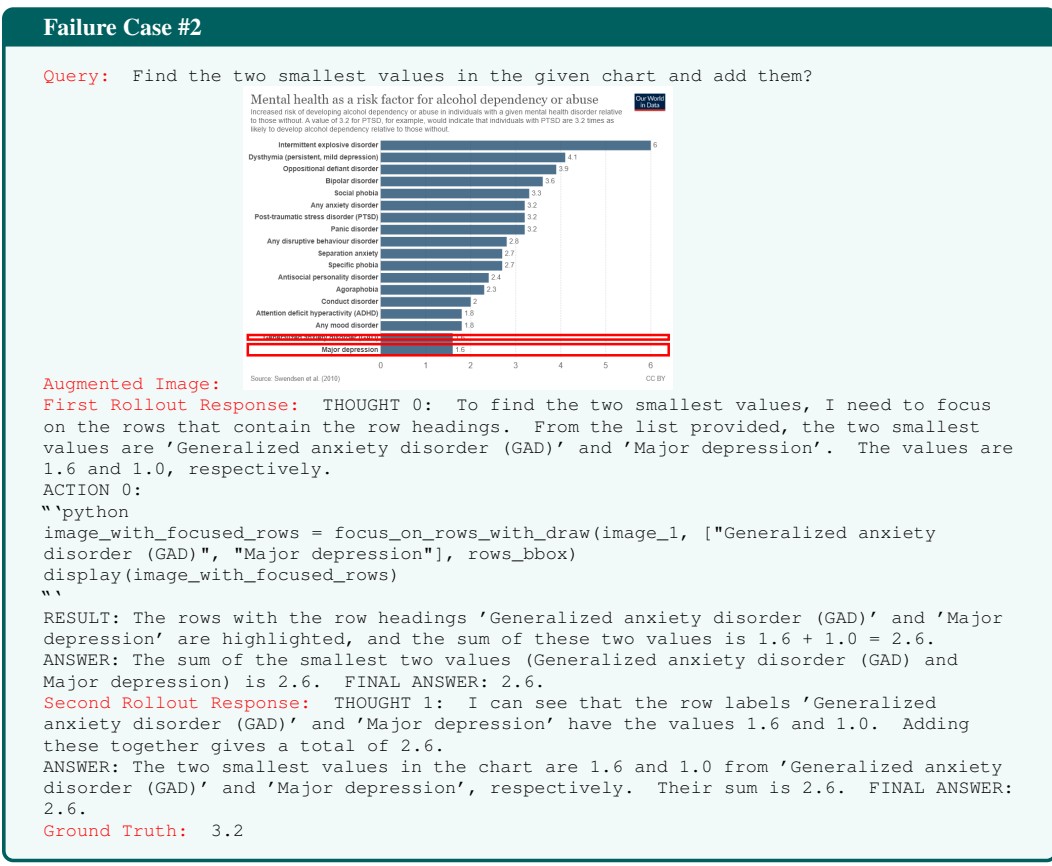

**Failure Case #2**

Query:  Find the two smallest values in the given chart and add them?

Augmented Image:
First Rollout Response:  THOUGHT 0:  To find the two smallest values, I need to focus on the rows that contain the row headings.  From the list provided, the two smallest values are 'Generalized anxiety disorder (GAD)' and 'Major depression'.  The values are 1.6 and 1.0, respectively.
ACTION 0:
```python
image_with_focused_rows = focus_on_rows_with_draw(image_1, ["Generalized anxiety disorder (GAD)", "Major depression"], rows_bbox)
display(image_with_focused_rows)
```
RESULT: The rows with the row headings 'Generalized anxiety disorder (GAD)' and 'Major depression' are highlighted, and the sum of these two values is 1.6 + 1.0 = 2.6.
ANSWER: The sum of the smallest two values (Generalized anxiety disorder (GAD) and Major depression) is 2.6.  FINAL ANSWER: 2.6.
Second Rollout Response:  THOUGHT 1:  I can see that the row labels 'Generalized anxiety disorder (GAD)' and 'Major depression' have the values 1.6 and 1.0.  Adding these together gives a total of 2.6.
ANSWER: The two smallest values in the chart are 1.6 and 1.0 from 'Generalized anxiety disorder (GAD)' and 'Major depression', respectively.  Their sum is 2.6.  FINAL ANSWER: 2.6.
Ground Truth:  3.2

3. The model concludes in the first rollout that augmentation is not needed, but fails to provided the correct answer.

**Failure Case #3**

Query: What is the percentage of the highest blue bar?

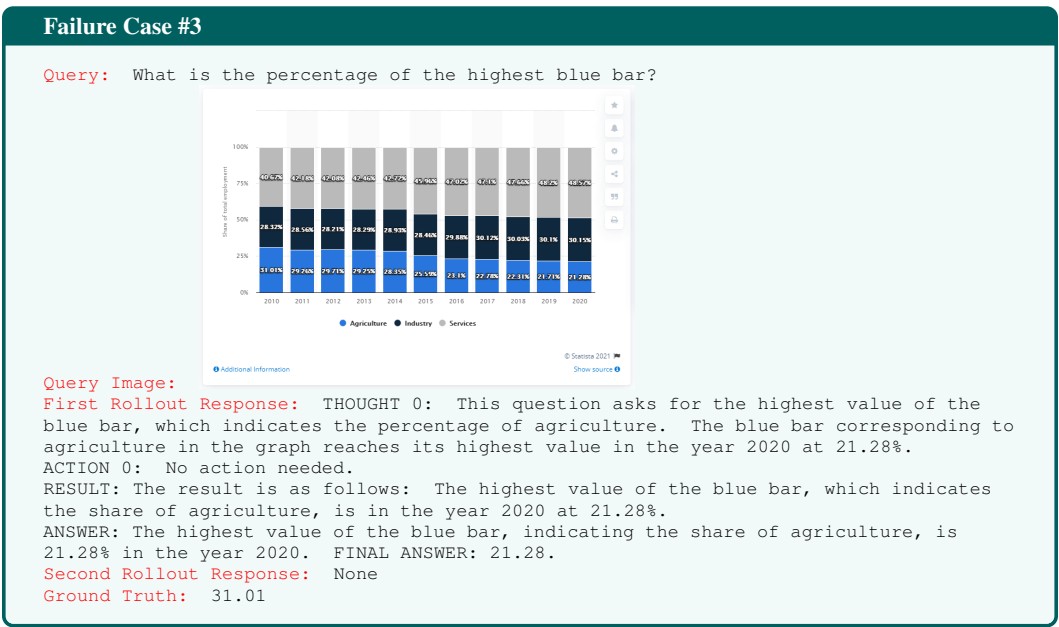

Query Image:
First Rollout Response: THOUGHT 0: This question asks for the highest value of the blue bar, which indicates the percentage of agriculture. The blue bar corresponding to agriculture in the graph reaches its highest value in the year 2020 at 21.28%.
ACTION 0: No action needed.
RESULT: The result is as follows: The highest value of the blue bar, which indicates the share of agriculture, is in the year 2020 at 21.28%.
ANSWER: The highest value of the blue bar, indicating the share of agriculture, is 21.28% in the year 2020. FINAL ANSWER: 21.28.
Second Rollout Response: None
Ground Truth: 31.01

4. The model generated code for augmentation but the code execution was unsuccessful.

**Failure Case #4**

Query: What is the average upper income class population?

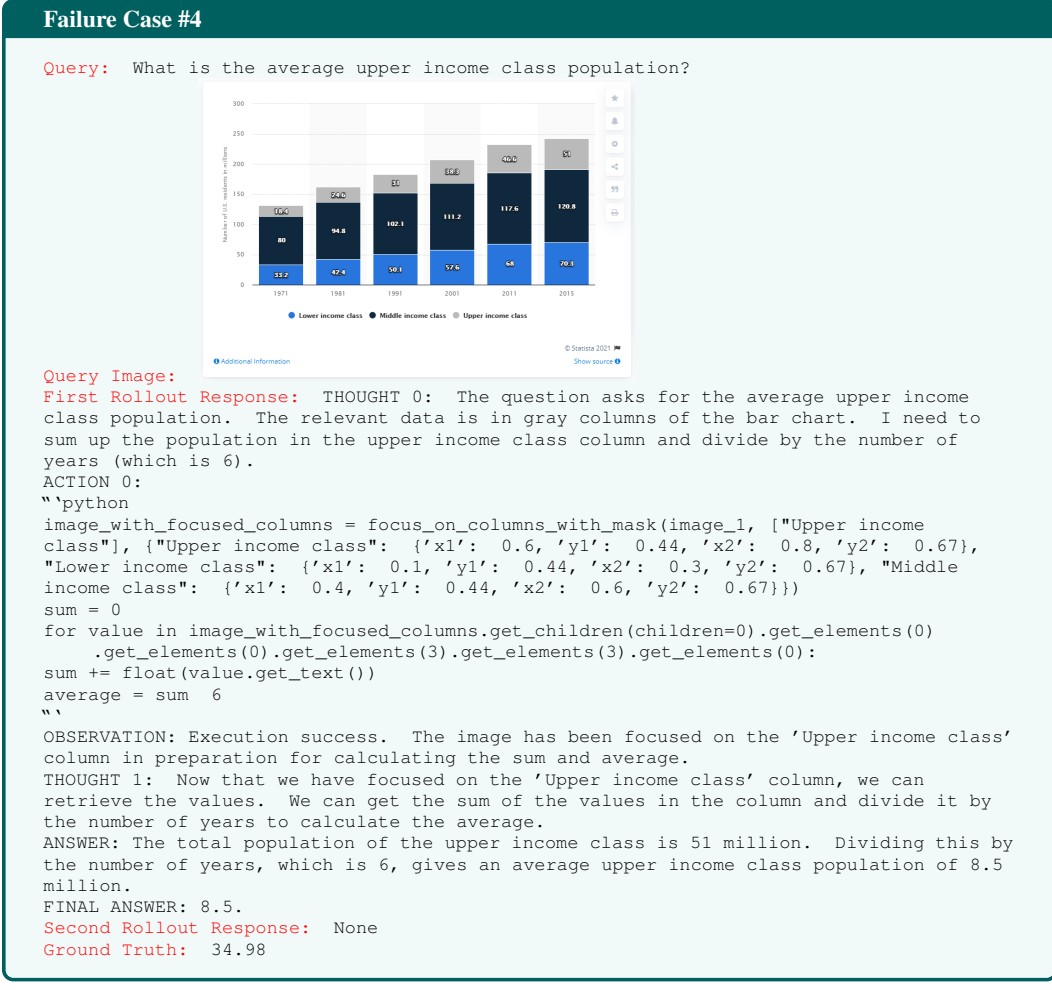

Query Image:
First Rollout Response: THOUGHT 0: The question asks for the average upper income class population. The relevant data is in gray columns of the bar chart. I need to sum up the population in the upper income class column and divide by the number of years (which is 6).
ACTION 0:
```python
image_with_focused_columns = focus_on_columns_with_mask(image_1, ["Upper income
class"], {"Upper income class": {'x1': 0.6, 'y1': 0.44, 'x2': 0.8, 'y2': 0.67},
"Lower income class": {'x1': 0.1, 'y1': 0.44, 'x2': 0.3, 'y2': 0.67}, "Middle
income class": {'x1': 0.4, 'y1': 0.44, 'x2': 0.6, 'y2': 0.67}})
sum = 0
for value in image_with_focused_columns.get_children(children=0).get_elements(0)
    .get_elements(0).get_elements(3).get_elements(3).get_elements(0):
sum += float(value.get_text())
average = sum  6
```
OBSERVATION: Execution success. The image has been focused on the 'Upper income class' column in preparation for calculating the sum and average.
THOUGHT 1: Now that we have focused on the 'Upper income class' column, we can retrieve the values. We can get the sum of the values in the column and divide it by the number of years to calculate the average.
ANSWER: The total population of the upper income class is 51 million. Dividing this by the number of years, which is 6, gives an average upper income class population of 8.5 million.
FINAL ANSWER: 8.5.
Second Rollout Response: None
Ground Truth: 34.98

