# OpenReview forum: "VTool-R1: VLMs Learn to Think with Images via Reinforcement Learning on Multimodal Tool Use"
_ICLR.cc/2026/Conference — ICLR 2026 Poster_

### Official Review · Reviewer_JqBn · 2025-10-24

**Soundness:** 2
**Presentation:** 2
**Contribution:** 2
**Rating:** 4
**Confidence:** 4

**Summary:**

This paper presents VTool-R1, a reinforcement learning finetuning (RFT) framework tailored for vision-language models (VLMs) that aims to teach these models to generate truly multimodal chains of thought by effectively interleaving text and intermediate visual reasoning steps using external Python-based visual editing tools. Unlike many recent RFT efforts which only incentivize text-based reasoning conditioned on images, VTool-R1 explicitly integrates visual tool invocation and allows VLMs to "think with images," optimizing for outcome-based rewards tied solely to task accuracy. The work provides comprehensive experimental results across structured visual question answering (VQA) benchmarks using both tables and charts, demonstrates outperforming or on-par tool-augmented reasoning compared to strong baselines, and analyzes dynamics of tool use during training.

**Strengths:**

1. The manuscript convincingly highlights the gap in existing VLMs' inability to reason with images in a stepwise, multimodal fashion—a limitation ignored by prior RFT works focusing on text-only COTs conditioned on static images.
2. The paper benchmarks across challenging settings (tables and charts), leverages SOTA VLMs (Qwen2.5-VL), and includes both accuracy results (Table 1) and detailed training dynamics (Figure 3), encompassing accuracy, tool call rates, and execution success. These support the claimed advances.
3. The VTool-R1 framework extends outcome-based RFT principles to visual tool-use, with an explicit two-stage rollout process and a policy gradient method (GRPO), as visualized in Figure 1. This design enables conditional tool invocation and reasoning directly over modified visual states.

**Weaknesses:**

1. The paper’s discussion of “tool use” is overly restricted to structured visual tasks (e.g., tables and charts), yet the title and abstract emphasize that the model “learns to think with images.” This framing creates an expectation of broader visual reasoning capabilities, but the actual scope is much narrower, leading to a mismatch between presentation and content. In the Related Work section, the review of prior research on tool-augmented VLMs and visual reasoning chains is present but lacks depth. In particular, the paper omits discussion of recent works on open-domain visual-language tool use, which would provide necessary context for positioning its contribution. Moreover, the paper does not analyze failure cases or qualitatively examine incorrect tool calls. Although the authors acknowledge that “tool use is not always beneficial for accuracy”, they do not provide representative failure examples. As a result, readers cannot assess when and why the proposed method fails. Finally, in the Conclusion, the authors claim that this work “is the first to teach VLMs intermediate visual reasoning steps via RFT”. However, this statement appears overstated—it gives the impression that the method is mature and widely applicable, whereas the actual experimental scope remains limited.
2. The authors claim that this work is the first reinforcement learning framework that enables vision-language models (VLMs) to learn through intermediate visual reasoning steps combined with tool use. However, the differences from prior work are not clearly articulated. For example, existing methods such as Visual Sketchpad (Hu et al., 2024) and OpenThinkIMG (Su et al., 2025) have already introduced visual sketching or editing steps during reasoning. The core idea—integrating tool invocation and visual editing into VLMs to enhance reasoning—is not fundamentally new in the tool-augmented or multimodal reasoning community. The paper lacks a deep analysis explaining why previous approaches are insufficient and why the proposed method works better. Furthermore, the authors emphasize using outcome-based rewards instead of process supervision. Yet, the absence of process-level guidance (i.e., when to invoke a tool and how to edit an image) may turn the method into a black-box fine-tuning procedure. The paper does not sufficiently justify why this design is preferable to process supervision.
3.The visual editing toolkit used in experiments is overly simplistic—limited to actions like highlighting, masking, or drawing boxes on rows and columns. These operations apply only to structured visual tasks such as tables and charts, yet the paper generalizes the approach as a universal paradigm for visual reasoning, which is a clear overreach from structured to real-world imagery. The datasets (VWTQ, VTabFact, ChartQA) are small-scale and relatively controlled. They do not reflect the diversity or complexity of open-domain multimodal reasoning, leaving generalization to broader settings unproven. The reward model relies on a lightweight LLM to judge answer correctness, which can be unreliable due to fuzzy matching, semantic variation, and inconsistent tool outputs. The authors themselves admit that tool-call correctness cannot be precisely evaluated, suggesting potential instability in the reward signal. Training only optimizes final answers, without explicitly supervising intermediate tool calls or visual edits. As a result, the model might simply learn to occasionally invoke a tool rather than to use tools meaningfully for reasoning, which undermines the core claim. There is a lack of thorough ablations: while Table 1 compares runs with and without tool calls, the paper does not analyze cases of tool misuse, tool failure, or scaling effects across model sizes (3B / 7B / 32B). Finally, the discussion of generalization and real-world applicability is weak. The authors acknowledge that their work focuses on structured visual reasoning, but this narrow scope significantly limits the claimed impact of the method.

**Questions:**

1. Can the authors provide baseline comparisons and/or qualitative ablations versus directly related, recent tool-use RL works like DeepEyes, ACTIVE-O3, MM-EUREKA, and DINO-R1? This is critical to position VTool-R1’s improvements in context.
2. Is there evidence that the proposed framework generalizes to more open-ended task domains, (e.g., beyond tables/charts) and to toolsets that provide more than highlighting/masking? Have any experiments been attempted in less-structured or multi-stage reasoning domains?
3. What was the measured reliability of the LLM-based reward judge? Was there a manual verification of reward assignment on a holdout set?

---

> ### Author Response · Authors · 2025-12-03
> **Response 1/2 to Reviewer JqBn**
>
> We appreciate reviewer JqBn’s comments, especially for acknowledging that our work tackles the important gap in existing VLMs' inability to reason with image chain of thoughts and provides strong experiments. Below, we provide detailed responses to the specific concerns and questions raised.
>
> **Q1: Concerns regarding two step design of VTool-R1’s tool use framework.**
>
> We focus on the two-step setup primarily for two reasons, as detailed in our response to Reviewer ajqX and Bc6R:
>
> (a) **Task structure and parallel tool design**. Similar to ReFocus [3], our tasks rely on visual editing tools that can naturally handle multiple operations in parallel within a single generated image step. For example, if a task requires highlighting two rows, or highlighting one region and masking another, the VLM can issue multiple tool calls in parallel and obtain a single edited image. In our experimental task setup, further breaking these operations into multiple sequential steps is not necessary to solve the tasks effectively.
>
> (b) **Current open-source VLM  limitations**. In early experiments with Qwen2.5-VL, we observed that its capability to robustly handle more than two input images is limited in complicated reasoning tasks, likely due to its pre-training regime. Simply increasing the number of image steps does not automatically yield better reasoning performance, especially when pre-training is not explicitly designed for heavily interleaved image–text interactions with many image steps on complex reasoning tasks.
>
> As future VLMs become stronger and truly multi-turn VL datasets emerge (imagine visual puzzle games that require the model to solve step by step), we plan to extend VTool-R1 beyond the current two-step design and explore more iterative tool-use reasoning strategies in the future.
>
> We also discuss additional **practical challenges of multi-turn VLM reasoning in terms of data, infra and model**, in our response (Q1 and A1) to reviewer Bc6R, please feel free to check it out if you are interested.
>
> **Q2: Questions on the contribution and novelty of VTool-R1**
>
> We acknowledge concurrent efforts and will revise our claim to "one of the first works" in this domain. Works like DeepEyes and OpenThinkIMG were completed around the same time as VTool-R1 (May 2025). As researchers, we are excited to see that these concurrent research efforts, while developed independently, approach the shared goal of **“thinking with images”** from different angles and are complementary in this emerging direction.
>
> However, VTool-R1 is fundamentally different from inference-only methods and text-only RFT mentioned in the author’s comments.
>
> Methods like Visual Sketchpad [6] and ReFocus [3] are fundamentally inference-only methods that lack training mechanisms and primarily benefit strong proprietary models. Smaller, open-sourced VLM models fail to generate executable code and thus benefit from intermediate tool-enabled visual steps without training. Our method outperforms strong VLM baselines trained using RL (R1-VL) without inter-leaving images and texts. VTool-R1 is one of the first works to successfully train VLMs to reason with interleaved images and texts..
>
> **Q3: More baselines**
>
> Regarding comparisons with recent tool-use RL works like DeepEyes [1], we note that DeepEyes is a concurrent work also under review in this ICLR cycle. Our method consistently demonstrates a clear advantage over concurrent strong baselines. Notably, DeepEyes 7B has an accuracy of 59.993% on the Chart Split of VTool-R1 given the python tools, while VTool-R1 achieved an 80.7% accuracy on the Chart Split. We want to highlight that the DeepEyes framework does not introduce a tool that allows VLMs to retrieve information via tool-use in an interleaved manner.
>
> **Q4: Why not process-level reward**
>
> Respectfully, we discussed the motivations as well as our experiments regarding process-level reward guidance, in the era of out-come based R1 style reward. We found that any attempt to incentivize models on a process-level with regards to tool use is prone to reward hacking in our tool use setup. Penalty for misuse of a tool, or rewarding the use of a tool, is easily exploited by the model. The VLM will learn to not use a tool at all, or make purposeless yet executable tool calls to earn rewards. Implementing fine-grained rewards also go against the general outcome-based reward practice in reinforcement learning. An outcome only based reward prevents reward hacking, and allows the model to implicitly learn strategic tool use. Regarding the black box argument, we respectfully disagree on this being a valid argument, since it can be used against all reinforcement learning methods and setups.

---

> ### Author Response · Authors · 2025-12-03
> **Response 2/2 to Reviewer JqBn**
>
> **Q5: Why lightweight LLM judge**
>
> We strictly use the same judging prompt and setup as ReFocus [3], and we manually inspected a subset of the dataset to verify its behavior. In these checks, the judge achieved an extremely high agreement rate with the human, which is good for semi–open-form VQA tasks.
>
> **Q6: Analysis on tool use behavior**
>
> Regarding the actual tool use performance of our model, we will also provide a detailed analysis of failure modes and corresponding discussion in the extra page of the final version.
>
> In particular, we highlight several representative categories of failure cases, including:
>
> (a). The model correctly generates an intermediate reasoning visual step, but fails to make the correct inference in the second rollout.
>
> (b). The model correctly identified the intermediate visual step needed for the reasoning, but the augmentation is slightly flawed (numbers obstructed by the added bounding box). The second rollout fails to read the correct numbers from the edited image.
>
> (c). The model concludes in the first rollout that no image reasoning step is not needed, but fails to provide the correct answer.
>
> (d). The model generated tool use code for but the code execution was unsuccessful, and no intermediate visual step is provided.
>
> **Q7: Extension to open-ended task domains**
>
> VTool-R1 does not impose a fundamental barrier to generalization; given suitable tasks and toolsets, it can be applied to open-ended settings. A key challenge for open-ended reasoning is the lack of benchmarks where multi-turn visual intermediate steps are actually necessary to solve the task. Many existing benchmarks (e.g., BLINK [4], MMMU [5]) can often be solved via direct visual perception or one-hop reasoning. Motivated by VTool-R1, we believe an important direction for future work is constructing genuinely multi-turn reasoning benchmarks, for which VTool-R1 provides a solid foundation towards.
>
> [1] DeepEyes: Incentivizing "Thinking with Images" via Reinforcement Learning, In submission to ICLR 2026, Submission ID 11632
>
> [2] OpenThinkIMG: Learning to Think with Images via Visual Tool Reinforcement Learning, arxiv, 2505.08617
>
> [3] ReFocus: Visual Editing as a Chain of Thought for Structured Image Understanding, ICML 2025
>
> [4] BLINK: Multimodal Large Language Models Can See but Not Perceive, ECCV 2024
>
> [5] MMMU: A Massive Multi-discipline Multimodal Understanding and Reasoning Benchmark for Expert AGI, CVPR 2024.
>
> [6] Visual Sketchpad: Sketching as a Visual Chain of Thought for Multimodal Language Models, NeurIPS 2024

---

### Official Review · Reviewer_4Mrt · 2025-10-31

**Soundness:** 3
**Presentation:** 3
**Contribution:** 2
**Rating:** 4
**Confidence:** 4

**Summary:**

This paper proposes VTool-R1, a reinforcement learning finetuning (RFT) framework for vision-language models (VLMs). The method aims to train VLMs to “think with images” — i.e., to generate multimodal chains of thought (CoT) by interleaving text reasoning and intermediate visual reasoning steps.
Unlike prior work that only uses images as static input, VTool-R1 integrates Python-based visual editing tools into the RFT loop, allowing models to modify images during reasoning. The approach employs outcome-based rewards rather than process supervision, trained with a Group Relative Policy Optimization (GRPO) objective adapted from DeepSeek-R1.
Experiments on structured visual reasoning tasks (tables, charts) demonstrate improvements in accuracy over both inference-only and non-tool-use baselines, using Qwen2.5-VL models (3B/7B/32B). The authors report that the model learns to selectively invoke visual tools to improve reasoning outcomes.

**Strengths:**

This paper presents a clear and technically solid framework that adapts reinforcement finetuning (RFT) to multimodal reasoning. I like that the authors focus on the gap between text-only reasoning and truly visual reasoning, and they provide a well-defined approach—integrating Python-based visual tools into the RL loop—to address it. The method is implemented cleanly and described in enough detail to be reproducible. The experiments, though limited in scope, are consistent and show that the model can indeed learn to use visual tools more strategically after RFT. Overall, the idea of teaching VLMs to “think with images” is appealing, timely, and potentially useful for future multimodal reasoning work.

**Weaknesses:**

From my perspective, the contribution feels somewhat incremental—the method mainly extends DeepSeek-R1–style RFT to VLMs without introducing new algorithmic ideas. The evaluation is narrow, focusing only on structured tasks like chart and table reasoning, which limits how convincing the results are for general multimodal reasoning. The reward design depends on an LLM-based judge, which is subjective, and the tool-use evaluation metric (simply checking if Python runs) doesn’t truly measure reasoning success. Some claims about being the “first” to enable multimodal chain-of-thought are overstated, given existing work such as Vision-R1 and Refocus.

**Questions:**

1. How does VTool-R1 generalize to multi-turn tool use or open-ended visual reasoning beyond structured data?

2. Can you provide human evaluation or ground-truth verification for visual tool correctness?

3. How sensitive is performance to reward signal noise (given LLM-judge subjectivity)?

4. What is the computational cost of two-stage inference and RFT rollouts relative to pure text-based RFT?

---

> ### Author Response · Authors · 2025-11-27
> **Response 1/3 to Reviewer 4Mrt**
>
> We appreciate reviewer 4Mrt’s insightful comments, especially for recognizing that our work presents a **clear and technically solid framework, with a well-defined approach, clean methodology, and consistent, effective experiments supporting an appealing, timely, and potentially useful idea for multimodal reasoning.** Taken together, we believe these strengths make VTool-R1 a valuable contribution to the community. Below, we provide detailed responses to the specific concerns and questions raised.
>
> **Q1: Concerns about incremental contribution, i.e., mainly extending RFT to VLMs with no algorithmic ideas in DeepSeek-R1-style.**
>
> A1: We respectfully disagree with the characterization that VTool-R1 is **mainly an extension of RFT** to VLMs without algorithmic novelty.
>
> While R1-style training is indeed part of the magical training recipe in our framework, the core contribution and novelty of VTool-R1 lies in **enabling VLMs to learn to reason with an interleaved visual–textual chain of thought, in a multi-turn manner**. All these happen with the dedicated design of visual editing tool use merging into RL rollout as well as inference and rollout stages, addressing fundamental limitation early inference-only works for multimodal chain of thoughts such as Visual Sketchpad [1], or ReFocus [2].
>
> Moreover, we would like to emphasize that “R1-style RFT extensions in VLM” have also been widely recognized as meaningful contributions in the community, with substantial algorithmic design space. For example, works such as R1-VL [3] and OpenVLThinker [4] directly adopt R1-style training, and prepare data for vision and language tasks. **VTool-R1 even steps further for advanced multi-turn, multi-modal reasoning capability beyond what these prior works explored.**
> We would be happy to further discuss this point if there are remaining concerns about the contribution of this line of work.
>
> **Q2: The claim of being the “first” to enable multimodal chains of thought is overstated, especially considering Vision-R1 and ReFocus.**
>
> A2: Thank you for the comment. We claim to be **one of the first works to train VLMs to generate multimodal chains of thought**. We give detailed reasons below why these works, Refocus [2] and works like concurrent Vision-R1 [5] are different from VTool-R1.
>
> Inference only works like Visual Sketchpad [1] and ReFocus [2] leverage intermediate visual steps, but they do not train VLMs to improve their own ability to generate such steps in the reasoning. As a result, they primarily demonstrate benefits for strong commercial models (e.g., GPT-level systems), while smaller open-source models struggle to produce meaningful intermediate visual edited steps. In particular, ReFocus, under a similar testing setup, reports that smaller open-source models cannot reliably generate useful intermediate visual edits, and instead they plug in GPT-generated intermediate images to see gains. In contrast, **VTool-R1 is the first to train VLMs to reasoning with multimodal chain of thoughts that interleaves images and texts**.
>
> These R1-style multimodal works such as Vision-R1 [5] and R1-VL [3], which we already compare against in our experiments, are fundamentally different from VTool-R1 in how “multimodality” appears in the reasoning process. **Their reasoning is carried out purely in text, conditioned on multimodal inputs**, whereas the intermediate steps in the chain of thought are textual only. In contrast, **VTool-R1 explicitly includes image intermediate steps in the chain of thought, in a multi-turn manner**, with the model learning when and how to invoke visual tools and to incorporate the resulting edited images into subsequent reasoning.
>
> We would be happy to further discuss this point if there are remaining concerns about the novelty of VTool-R1.

---

> ### Author Response · Authors · 2025-11-27
> **Response 2/3 to Reviewer 4Mrt**
>
> **Q3. How does the VTool-R1 generalizes to multi-turn tool use or handle more open-ended visual reasoning tasks?**
>
> A3: Thank you for the insightful question! VTool-R1 focuses on the two-step setup primarily for two reasons:
>
> (a) **Task structure and parallel tool design**. Similar to ReFocus [2], our tasks rely on visual editing tools that can naturally handle multiple operations in parallel within a single generated image step. For example, if a task requires highlighting two rows, or highlighting one region and masking another, the VLM can issue multiple tool calls in parallel and obtain a single edited image. In our experimental task setup, further breaking these operations into multiple sequential steps is not necessary to solve the tasks effectively.
>
> (b) **Current open-source VLM  limitations**. In early experiments with Qwen2.5-VL, we observed that its capability to robustly handle more than two input images is limited in complicated reasoning tasks, likely due to its pre-training regime. Simply increasing the number of image steps does not automatically yield better reasoning performance, especially when pre-training is not explicitly designed for heavily interleaved image–text interactions with many image steps on complex reasoning tasks.
>
> As future VLMs become stronger and truly multi-turn VL datasets emerge (imagine visual puzzle games that require the model to solve step by step), we plan to extend VTool-R1 beyond the current two-step design and explore more iterative tool-use strategies in the future
>
> We also discuss additional **practical challenges of multi-turn VLM reasoning** in our response (Q1 and A1) to reviewer Bc6R, please feel free to check it out if you are interested.
>
> Regarding open-ended visual reasoning, our framework does not impose a fundamental barrier to generalization: given suitable tasks and toolsets, VTool-R1 can, in principle, be applied to more open-ended settings. **A key open question, however, is when multi-turn visual intermediate steps are actually necessary**. Many existing more open-ended benchmarks, such as single-hop or vision-centric questions in Blink [6], or knowledge-heavy benchmarks like MMMU [7], can often be solved without real multi-turn reasoning. Motivated by VTool-R1, an important direction for future work is to design and **construct genuinely multi-turn reasoning multimodal benchmarks where sequential tool use and intermediate visual states are required to solve the task**, and then evaluate and extend our framework in those settings.
>
> **Q4. The reward design depends on a subjective LLM judge. How sensitive is the training to reward signal noise?**
>
> A4: Thank you for the feedback. We would like to clarify that our LLM judge is not **purely subjective**. The judge only **compares the model’s final prediction against the ground-truth answer**, rather than providing free-form subjective evaluation. We strictly use the same judging prompt and setup as ReFocus [2], and we manually inspected a subset of the dataset to verify its behavior. In these checks, the judge achieved an extremely high agreement rate with the human, which is good for semi–open-form VQA tasks.
>
> Empirically, we observe that the reward noise during training is quite low, comparable to verifiable tasks that rely on Python-based verifiers like mathruler, which can also occasionally misjudge answers.
>
> We will include the detailed judge prompt and explicitly clarify in the appendix that the judge is used purely for matching between ground-truth answers and model predictions. Thank you for suggesting us to make this clearer in the manuscript.

---

> ### Author Response · Authors · 2025-11-27
> **Response 3/3 to Reviewer 4Mrt**
>
> **Q5, Tool-use success evaluation criteria, any human evaluation?**
>
> A5: Thank you for raising this point. This is an interesting detail in our paper.
>
> In the paper, we report tool-use rates and explicitly note that human evaluation is necessary for obtaining accurate estimates. For the automatic tool-use metric in our figure, we define a “successful” tool-use episode as one where the model **(i) executes the generated Python code without error, (ii) runs to the final display() call and successfully generates an intermediate image step\, and (iii) invokes at least one tool from the predefined toolset.**
>
> To better understand its reliability, we conducted a human evaluation on a subset of 100 traces from the 3B model trained with VTool-R1 in the chart. Under the automatic criterion, the tool-use success rate is 45%, whereas human annotators judged only 39% of these cases as genuinely successful. The discrepancy largely comes from cases where the model hallucinates tool inputs, invokes tools outside the defined toolset, or calls dummy tools that do not meaningfully contribute to solving the task.
>
> This gap reinforces a claim in our paper: **without robust criteria for determining tool-call success, introducing tool-call–based rewards risks reward hacking.** For this reason, VTool-R1 deliberately focuses on outcome-based rewards (via answer correctness), which we believe is a **safer and more stable choice** in the early exploration current tool-use setting in VLM.
>
> **Q6. What is the computational cost of two-stage inference and RFT rollouts compared to standard text-only approaches?**
>
> A6: The overhead largely depends on implementation details and hardware.
>
> In our initial implementation (around May 2025), two-stage inference was noticeably slower, over 30% slower than single-stage inference in wall-clock time, because the GPU sat idle while waiting for tool calls to complete and new images to be generated.
>
> In our recent implementation, we adopt a multi-turn asynchronous inference pipeline based on the agent loop (introduced around Aug 2025) in the open-source VeRL framework [8]. With this design, tool calls and image generation are better overlapped with model computation, so the wall-clock inference time between single-turn and multi-turn setups becomes much closer (less than 10%), with only modest overhead in practice and not much GPU idle time.
>
> **Ending Remark:**
>
> Thank you again for your time and expertise, and for the constructive feedback. We sincerely hope our responses address your concerns and clarify the contributions of VTool-R1. Given that the paper currently has three borderline-reject scores and one accept, your strong support and raising score would be greatly appreciated if you find that VTool-R1 opens new opportunities for RL-based multi-turn VLM reasoning with multimodal chains of thought. We look forward to further discussion as well.
>
> [1] Visual Sketchpad: Sketching as a Visual Chain of Thought for Multimodal Language Models, NeurIPS 2024
>
> [2] ReFocus: Visual Editing as a Chain of Thought for Structured Image Understanding, ICML 2024.
>
> [3] R1-VL: Learning to Reason with Multimodal Large Language Models via Step-wise Group Relative Policy Optimization, ICCV 2025
>
> [4] OpenVLThinker: Complex Vision-Language Reasoning via Iterative SFT-RL Cycles, NeurIPS 2025.
>
> [5] Vision-R1: Incentivizing Reasoning Capability in Multimodal Large Language Model, ICLR 2026 Submission ID: 4366.
>
> [6] BLINK: Multimodal Large Language Models Can See but Not Perceive, ECCV 2024
>
> [7] MMMU: A Massive Multi-discipline Multimodal Understanding and Reasoning Benchmark for Expert AGI, CVPR 2024.
>
> [8] VeRL: Volcano Engine Reinforcement Learning for LLMs, Github, Bytedance Seed.

---

### Official Review · Reviewer_Bc6R · 2025-10-31

**Soundness:** 4
**Presentation:** 4
**Contribution:** 4
**Rating:** 8
**Confidence:** 4

**Summary:**

The paper introduces VTool-R1, a framework that applies reinforcement learning finetuning (RFT) to train vision-language models (VLMs) to generate multimodal chains of thought by interleaving text and intermediate visual reasoning steps. Unlike prior work, which focuses solely on textual reasoning given images, VTool-R1 leverages Python-based visual editing tools during training, allowing the model to learn how and when to use visual steps that can improve structured visual reasoning tasks. The framework uses outcome-based rewards to incentivize improved reasoning. Extensive experiments on chart and table-based reasoning tasks show that VTool-R1 enables VLMs to “think with images”—improving performance, especially with open-source models previously unable to use tools meaningfully.

**Strengths:**

1. VTool-R1 is the first research work to demonstrate RL can train VLMs to interleave visual steps within a chain of thought using python-based visual editing tools.
2. Strong, controlled experiments and comparisons with state-of-the-art baselines, including commercial and open-source models, show clear improvement and robust methodology.
3. The framework and training design have potential to generalize to more diverse tools and reasoning tasks.
4. Explanations are accessible, with step-by-step rationale for design choices, experimental setup, and limitations.

**Weaknesses:**

1. Single-Turn Tool Use: The model is only trained/tested with one round of tool invocation, limiting application in multi-step, interactive reasoning.
2. Limited Task Scope: The experiments focus only on structured chart and table reasoning with a simple, small, predefined set of visual editing tools.
3. Lack of Extensive Qualitative Comparison: The paper would benefit from further qualitative comparison.

**Questions:**

1.  How does VTool-R1 perform when applied to less-structured images (e.g., photographs or scenes)?
2. What are the key challenges to scaling the VTool-R1 approach to multi-turn, iterative tool use?
3. Can you provide more detailed behavioral analysis of failure modes? eg. will the model misuse some tool?

---

> ### Author Response · Authors · 2025-11-27
> **Response 1/2 to Reviewer Bc6R**
>
> We appreciate reviewer Bc6R’s insightful comments and strong support of VTool-R1, especially for recognizing the **novelty of our work in using RL to train VLMs to interleave visual steps within chains of thought**. We also thank the reviewer for acknowledging the **strength of our strong experimental results and the clarity of our explanations**. Below, we provide detailed responses to address the specific concerns and questions raised.
>
> **Q1: Why does the paper focus on single-turn tool use? What are the key challenges in extending to multi-turn tool use?**
>
> A1: In VTool-R1, we focus on the two-step setup primarily for two reasons:
>
> (a) **Task structure and parallel tool design**. Similar to ReFocus [1], our tasks rely on visual editing tools that can naturally handle multiple operations in parallel within a single generated image step. For example, if a task requires highlighting two rows, or highlighting one region and masking another, the VLM can issue multiple tool calls in parallel and obtain a single edited image. In our experimental task setup, further breaking these operations into multiple sequential steps is not necessary to solve the tasks effectively.
>
> (b) **Current open-source VLM limitations.** In early experiments with Qwen2.5-VL, we observed that its capability to robustly handle more than two input images is limited in complicated reasoning tasks, likely due to its pre-training regime. Simply increasing the number of image steps does not automatically yield better reasoning performance, especially when pre-training is not explicitly designed for heavily interleaved image–text interactions with many image steps on complex reasoning tasks.
>
> The current two-step setup in VTool-R1 could be viewed as a novel proof of concept that VLMs can interleave visual and textual chains of thought via RL, rather than as the endpoint for the community. We fully agree that real multi-turn VLM reasoning, and agentic behaviors are in the road map, and it faces several important challenges:
>
> (a) **Lack of multi-turn VLM reasoning datasets with sequential tool use.**
>  To the best of our knowledge, there is currently no benchmark in the vision language domain that systematically evaluates multi-turn tool use with truly sequential image-based intermediate steps or tool use. Ideally, such a benchmark would contain tasks that require a sequence of tool calls and multiple intermediate image states for subsequent reasoning (e.g., visual puzzle tasks that must be solved step by step, with sequential reasoning dependency).
>
> (b) **Training Infrastructure challenges.**
>  Multi-turn RL introduces new demands on RL training infrastructure. Interacting with tools over multiple rounds to generate images, and handling rollout trajectories with highly variable lengths within a batch requires specialized, asynchronous RL training systems that can support multi-turn, multimodal rollouts without incurring excessive GPU idle time. As far as we know, open-source RL infrastructures that natively support multi-turn, multi-modal RL are still very rare, with early and not-yet-perfect support (starting from ~08/2025)  emerging from open-source frameworks such as VeRL [2].
>
> (c) **Model challenges: context window length and multi-image capability.**
>  As noted above, current open-source VLMs often experience accuracy degradation when many images are included in the prompt for complex reasoning tasks. In addition, their context windows are limited, and images consume many more tokens than text. This makes it difficult to perform truly free-form multi-turn visual interaction without aggressive downscaling or cropping (for example, heavily cropped images in concurrent industrial work DeepEyes [3]).
>
> We see VTool-R1 as a step toward addressing these challenges by demonstrating that RL can successfully train VLMs to “think with images,” while also highlighting concrete directions and obstacles for future work on fully multi-turn multimodal reasoning. Also we hope VTool-R1 can motivate more work in the broader context of multi-turn and multimodal reasoning in the future.

---

> ### Author Response · Authors · 2025-11-27
> **Response 2/2 to Reviewer Bc6R**
>
> **Q2:  Limited tasks scope on structured table and chart, with pre-fixed toolset: How does VTool-R1 perform when applied to less-structured images (such as photographs or scenes)**
>
> A2: Thank you for your inspiring comments, VTool-R1 does not impose a fundamental barrier to generalization to open ended visual reasoning: given suitable tasks and well-defined toolsets, VTool-R1 can, in principle, be applied to more open-ended vision and language reasoning settings.
>
> **A key open question, however, is when multi-turn visual intermediate steps are actually necessary**. Many existing more open-ended benchmarks, such as single-hop or vision-centric questions in Blink [4], or knowledge-heavy benchmarks like MMMU [5], can often be solved without real multi-turn reasoning, and it is hard to define a meaningful toolset for these datasets.
>
> Our current work focuses on structured tables and charts because reasoning tasks in these domains align particularly well with the visual editing toolset (e.g., row/column highlighting, masking, region selection) originally developed in ReFocus [1]. This alignment makes them an intuitive and clean testbed for studying the effect of interleaving visual and textual chains of thought: the tools provide highly selective visual attention over structured images, which is ideal for showcasing the benefit of intermediate visual steps in reasoning.
>
> Extending VTool-R1 to less structured images would require designing and training new toolsets tailored to reasoning tasks in those domains. For example:
>
> In photographic scenarios, tools could resemble Photoshop-style editing tool sets (from a designer’s perspective) or user-oriented tools for answering multi-step visual questions,depending on downstream reasoning needs.
>
> For complex scenes, tools might focus on object retrieval, anomaly or “unnatural” object labeling, object heatmaps, or even integration with robotics-oriented reasoning tasks.
>
> We view these as promising directions inspired by VTool-R1, but building the necessary multi-turn reasoning datasets and toolsets, and retraining models accordingly, requires substantial effort in reasoning data curation, tool design, and RL training, which is beyond what can be accomplished during the short rebuttal period, with academic resource. We are happy to keep pushing these interesting ideas forward.
>
> **Q3: Lack of extensive qualitative comparison, any detailed behaviour analysis of failure modes?**
>
> A3: Thank you for the questions. We will provide a detailed analysis of failure modes and corresponding discussion in the extra page of the final version.
>
> In particular, we highlight several representative categories of failure cases, including:
>
> (a). The model correctly generates an intermediate reasoning visual step, but fails to make the correct inference in the second rollout.
>
> (b). The model correctly identified the intermediate visual step needed for the reasoning, but the augmentation is slightly flawed (numbers obstructed by the added bounding box). The second rollout fails to read the correct numbers from the edited image.
>
> (c). The model concludes in the first rollout that no image reasoning step is not needed, but fails to provide the correct answer.
>
> (d). The model generated tool use code for but the code execution was unsuccessful, and no intermediate visual step is provided.
>
> **Ending Remark:**
>
> Thank you again for your time and expertise. Given that the paper currently has three borderline-reject scores and one accept, we would be very grateful for your continued support during the final discussion phase if you feel our responses address your concerns. We look forward to further discussion and to exploring the new opportunities that VTool-R1 opens for RL-based interleaved image–text reasoning.
>
> [1] ReFocus: Visual Editing as a Chain of Thought for Structured Image Understanding, ICML 2025
>
> [2] VeRL: Volcano Engine Reinforcement Learning for LLMs, Github, Bytedance Seed.
>
> [3] DeepEyes: Incentivizing "Thinking with Images" via Reinforcement Learning, In submission to ICLR 2026, Submission ID 11632
>
> [4] OpenVLThinker: Complex Vision-Language Reasoning via Iterative SFT-RL Cycles, NeurIPS 2025.
>
> [5] Vision-R1: Incentivizing Reasoning Capability in Multimodal Large Language Model, ICLR 2026 Submission ID: 4366.

---

### Official Review · Reviewer_ajqX · 2025-11-01

**Soundness:** 2
**Presentation:** 2
**Contribution:** 2
**Rating:** 4
**Confidence:** 4

**Summary:**

This paper proposes an RFT (Reinforcement Fine-Tuning) framework that supports VLM multimodal reasoning with external visual editing tool usage. It defines multimodal reasoning as a two-step rollout. The experiment in the table and chart image scenes shows that the RFT training brings improvements for baseline models.

**Strengths:**

1. This work extends the textual reasoning in multimodal understanding to the multimodal reasoning that involves images and text.
2. This work successfully enables VLMs to learn to integrate intermediate visual reasoning steps into text-based chains of thought in the generated response.

**Weaknesses:**

1. It is not recommended to use expressions like "the first RFT framework that trains VLMs to generate multimodal chains of thought". I think the expression "first" is questionable.
2. As shown in Figure 1, can only two-step Reasoning be performed during model training? What about during inference after model training? Why wasn't it designed as a reasoning process with a maximum number of rounds? This is more in line with the phenomenon of multiple iterations in multimodal reasoning.
3. The experiment was insufficient. The experiment was conducted only on the two data Settings of Table Split and Chart Split, and it was only an in-domain experiment. The baseline models and benchmarks covered in the experiment are insufficient, which fails to reflect the effectiveness and universality of this method.
4. In the results of Table 1 and Table 2, why is there a situation where the result of "Tool Use" is worse than that of "Pure Run"? For example, in Qwen2.5-VL-32B and GPT-4o.

**Questions:**

1. Refer to the issues raised in the weakness section.
2. The curve of the model training process shown in Figure 3 has the risk of overfitting. A significant performance improvement was achieved after only 50 training steps. How many steps are generally trained for the model in this article?

---

> ### Author Response · Authors · 2025-11-27
> **Response 1/3 to Reviewer ajqX**
>
> We thank reviewer ajqX for the insightful comments and for highlighting our paper’s contributions in **extending textual reasoning to the multimodal setting** and enabling VLMs to learn to **integrate intermediate visual reasoning steps into text-based chain of thoughts**. Below, we provide a detailed response to each concern.
>
> **Q1: Questionable claim of being “the first work” in RFT that trains VLMs to generate multimodal chains of thought.**
>
> A1: Thank you for pointing this out. We will revise our claim to **“one of the first works”** in this domain, acknowledging concurrent efforts that also explore RFT for generating multimodal chains of thought. In particular, we will cite recent concurrent works including DeepEyes [1] and OpenThinkIMG [2] in the final version of the paper. **These two works and our VTool-R1 were all first made public in May 2025**. Notably, DeepEyes is also under review in the same ICLR cycle (submission ID 11632) and its reviews recognize its novelty.
>
> **We would like to emphasize that the existence of concurrent work in this general direction does not diminish the novelty of VTool-R1.** As researchers, we are excited to see that these concurrent research efforts, while developed independently, approach the shared goal of **“thinking with images”** from different angles and are complementary in this emerging direction.
>
> **Q2: Why limited to two-step reasoning? Can it be more iterative with a maximum number of rounds?**
>
> A2: This is a very insightful question and touches on exactly where the community is heading for, **toward multi-turn, iterative and interleaved reasoning in VLMs with RFT**. In the current version of VTool-R1, we adopt a two-step reasoning process with one intermediate visual step, **as proof of concept of “interleaving strategy works!”**. During both training and inference, the model has the flexibility to decide whether or not to invoke this intermediate visual reasoning step.
>
> We focus on the two-step setup primarily for two reasons:
>
> (a) **Task structure and parallel tool design**. Similar to ReFocus [3], our tasks rely on visual editing tools that can naturally handle multiple operations in parallel within a single generated image step. For example, if a task requires highlighting two rows, or highlighting one region and masking another, the VLM can issue multiple tool calls in parallel and obtain a single edited image. In our experimental task setup, further breaking these operations into multiple sequential steps is not necessary to solve the tasks effectively.
>
> (b) **Current open-source VLM  limitations**. In early experiments with Qwen2.5-VL, we observed that its capability to robustly handle more than two input images is limited in complicated reasoning tasks, likely due to its pre-training regime. Simply increasing the number of image steps does not automatically yield better reasoning performance, especially when pre-training is not explicitly designed for heavily interleaved image–text interactions with many image steps on complex reasoning tasks.
>
> As future VLMs become stronger and truly multi-turn VL datasets emerge (imagine visual puzzle games that require the model to solve step by step), we plan to extend VTool-R1 beyond the current two-step design and explore more iterative tool-use reasoning strategies in the future.
>
> We also discuss additional **practical challenges of multi-turn VLM reasoning in terms of data, infra and model**, in our response (Q1 and A1) to reviewer Bc6R, please feel free to check it out if you are interested. Looking forward to more discussion here.

---

> ### Author Response · Authors · 2025-11-27
> **Response 2/3 to Reviewer ajqX**
>
> **Q3: Concerns about Insufficient experiments, only in two data settings of Table and Chart, and it was in domain. Less baseline models and benchmarks covered.**
>
> A3: We are happy to provide additional out-of-domain experiments on table and chart tasks and to report more RL-trained baselines in this rebuttal.
>
> To address generalization, we conduct out-of-domain evaluations by training on chart-based tool use with horizontal bar plots and testing on vertical bar plots. These experiments show that the model’s learned tool-use behavior and multimodal chain-of-thought reasoning can generalize beyond a narrow domain, as long as the similar tool sets are well aligned with the task:
>
> |   Out-of-Domain| Tool Use Before Training| VTool-R1|
> |---------------------|---------------:|---------------:|
> | 3B    |       26.5     |      54.0      |
> | 7B   |       50.8   |    76.4   |
>
> Additionally, we include new results comparing our approach against a concurrent RL-trained tool-use VLM, DeepEyes [1]. We directly download their released 7B checkpoint and evaluate it on our chart split with our toolsets. Our 7B model trained with VTool-R1 achieves higher performance:
>
> |  Tool-use RFT Model | Deepeye[1] 7B| VTool-R1 7B|
> |---------------------|---------------:|---------------:|
> | Chart (best)   |       60.0     |      80.7      |
>
> Overall, our method consistently demonstrates a clear advantage over these concurrent strong baselines, further validating the effectiveness of our tool-augmented RL fine-tuning approach. **VTool-R1 is a strong post-training strategy, especially when we have a specialized toolset designed for multi-step visual reasoning tasks.**
>
> Regarding broader data settings:
>
> We agree it is valuable to extend VTool-R1 beyond table and chart tasks. However, designing new datasets requires substantial effort in data selection, specialized tool-set design, and model training, which cannot be completed within the short discussion period. We view our current experiments in structured image as a strong proof of concept that VLMs can learn to use visual tools and incorporate them as intermediate visual steps in a RL framework, and we plan to explore additional domains in future work.
>
> We have also added more discussion on why structured-image experiments are the most natural starting point for this proof of concept, along with envisioned extensions to broader settings, in our response (Q2 and A2) to reviewer Bc6R; please feel free to refer to that if you are interested: **in our view, structured images are the most natural and intuitive starting point for this line of work.**
>
> We will add the new experiment results in the final version of the paper. Thank you for your constructive feedback to the manuscript!
>
> **Q4: Why are some situations where tool use is worse than pure run, (such as 32B and GPT4-o)**
>
> A4: For smaller models (3B and 7B), before RFT, these models have relatively weak tool-use capabilities, leading to noticeably worse performance with tools than in pure reasoning mode. After applying our VTool-R1 RFT procedure, their tool-use performance improves dramatically and surpasses both of tool-use and pure-run (stronger) performance before our training.
>
> For more capable models like Qwen-2.5 VL 32B, the situation differs. These models already demonstrate reasonably strong tool-use capabilities before RFT, and their **pure reasoning accuracy is already extremely high (80%+)**. In this regime, additional tool use instructions can risk slightly degrading their performance because we are teaching the models to do what they have not learned in larger scale pre-training. As a result, even though our RFT can still improve 32B’s tool-use performance compared to its pre-training tool use, the pure-run accuracy has been so strong that additional gains become difficult, unless we have larger scale of training and more dedicated data design. (VTool-R1 is purely done with academic resources, our compute budget does not allow extensive RFT experiments at 32B scale, and we will explore this in the future as our computing scales.)
>
> For GPT-4o, we can only perform zero-shot evaluations. We demonstrate that even for such strong proprietary VLMs, **zero-shot tool use and incorporating intermediate visual steps remains challenging**. This further underscores the value of our method as a post-training technique that explicitly teaches models to reason with visual tools and intermediate visual steps.
>
> We will add a paragraph following up this discussion in the final version of the paper. Thank you for your constructive feedback to the manuscript!

---

> ### Author Response · Authors · 2025-11-27
> **Response 3/3 to Reviewer ajqX**
>
> **Q5 Concerns about overfitting of training in the training figure?**
>
> A5: In the reported figure, the batch size for weight updates is 256, and the training split contains 14,344 samples, resulting in roughly 56 steps per epoch. In RFT-on-VLM work such as OpenVLThinker [4], **training for a single epoch is standard practice**, and RFT is known to yield rapid performance improvements in this regime, as shown by recent works [4,5] in the literature.
>
> We do not observe evidence of overfitting in our validation curves; however, we would be happy to provide additional training details or plots if the reviewer has specific concerns regarding overfitting
>
> **Ending Remark:**
>
> Thank you again for your time and expertise, and for the constructive feedback. We sincerely hope our responses address your concerns and clarify the contributions of VTool-R1. Given that the paper currently has three borderline-reject scores and one accept, your strong support and raising score would be greatly appreciated if you find that VTool-R1 opens new opportunities for RL-based multi-turn VLM reasoning with multimodal chains of thought. We look forward to further discussion as well.
>
> [1] DeepEyes: Incentivizing "Thinking with Images" via Reinforcement Learning, In submission to ICLR 2026, Submission ID 11632
>
> [2] OpenThinkIMG: Learning to Think with Images via Visual Tool Reinforcement Learning, arxiv, 2505.08617
>
> [3] ReFocus: Visual Editing as a Chain of Thought for Structured Image Understanding, ICML 2025
>
> [4] OpenVLThinker: Complex Vision-Language Reasoning via Iterative SFT-RL Cycles, Neurips 2025,
>
> [5] VL-Rethinker: Incentivizing Self-Reflection of Vision-Language Models with Reinforcement Learning, Neurips 2025

---

### Author Response · Authors · 2025-12-03
**Brief Summary of Response**

We would like to thank the reviewers, AC, and SAC for their time and expertise throughout this reviewing process, especially the significant efforts from new ACs during this challenging period for open science. Below, we summarize the rebuttal and discussion process.

**Discussion Timeline**

Before the reviewer information leakage, we submitted detailed rebuttals to reviewers ajqX, Bc6R, and 4Mrt, and filed a confidential message reporting the AI-generated review JqBn, providing detailed evidence of **hallucinated and misused references, false claims, and two semantically overlapping weakness paragraphs that appear to be two duplicate AI-generated passes.**

After the information leakage, we also posted a rebuttal addressing some of the issues in the AI-generated review JqBn, but it was not possible to respond to every point due to the large number of hallucinations and incorrect claims.

**Discussion Summary**

Our work received broad recognition for **novelty in successfully training VLMs to integrate intermediate visual chains of thought via RFT** (ajqX, Bc6R, 4Mrt), **strong, controlled, and consistent experimental design** (Bc6R, 4Mrt), **timely and appealing idea with generalizable potential and broader impact** (Bc6R, 4Mrt),  **clear writing and explanations** (Bc6R).

We did not receive any responses from the reviewers and thus lost the opportunity for further discussion due to the OpenReview leakage issue. **However, we believe our rebuttal thoroughly addressed the main concerns**, including:

**First work or not** (ajqX Q1/A1, 4Mrt Q2/A2)

 We argue that our work is the first to train VLMs to perform multimodal chains of thought with visual tool use, while explicitly acknowledging concurrent works released in a similar timeframe that approach related problems from different, complementary angles. Together, these works are supplementary to each other in the emerging domain of “thinking with images.”

**Two-step vs. multi-turn visual reasoning steps** (ajqX Q2/A2, Bc6R Q1/A1, 4Mrt Q3/A3)

 We clarify that VTool-R1 naturally generalizes beyond two-step setups to multi-step visual reasoning, and we call for more multi-turn visual reasoning benchmarks, better infrastructure support, and stronger base models in the future.

**More baselines, more tasks, broader settings** (ajqX Q3/A3, Bc6R Q2/A2, 4Mrt Q3/A3)

 We fully address these comments by adding the new RL-trained baseline DeepEyes, introducing new out-of-domain experiments, and demonstrating the potential of VTool-R1 to generalize to broader tasks. We also note that current benchmarks still lack realistic multi-turn visual reasoning settings, which limits what can be evaluated.

**Failure analysis on tools and results** (Bc6R Q3/A3, 4Mrt Q5/A5)

 We include additional qualitative failure analyses and a human evaluation of tool-call success to address concerns about where and how the method/tool use fails.

**Sensitivity of the LLM judge to the answers** (4Mrt Q4/A4)

We show that our LLM-based judge is robust for semi-open-form VQA and matches ground truth with prediction well.

We also clarify further on extra questions raised by reviewer, including special case tool use performing worse (ajqX Q4/A4), overfitting or not (ajqX Q5/A5), contribution clarification (4Mrt Q1/A1), computational cost (4Mrt Q6/A6).

Unfortunately, the reviewers could not further engage with our rebuttal or update their scores due to the unexpected leakage issue, and we sincerely hope the ACs can take this context into consideration. Also, we do not include the **AI-generated review JqBn** in this discussion summary.

---

### Meta-Review · Area_Chair_uRsz · 2026-01-07

**Summary:**

This work introduces VTool-R1, a reinforcement learning finetuning (RFT) framework for vision-language models (VLMs). The approach aims to train VLMs to “think with images” (to generate multimodal chains of thought (CoT) by interleaving text reasoning and intermediate visual reasoning steps). This method extends the textual reasoning in multimodal understanding to the multimodal reasoning that involves images and text. It successfully enables VLMs to learn to integrate intermediate visual reasoning steps into text-based chains of thought in the generated response. Reviewers have pointed out some experiments are missing, which seem to be well addressed in rebuttal.

**Reviewer Concerns:**

Reviewers have pointed out some experiments are missing, which seem to be well addressed in rebuttal.

**Reviewer Scores:**

Reviewers are likely to increase the rating.

---

### Decision · Program_Chairs · 2026-01-26

Accept (Poster)